# Transient CAR T cells with specificity to oncofetal glycosaminoglycans in solid tumors

Nastaran Khazamipour[1,2], Htoo Zarni Oo [1,2,3,9], Nader Al-Nakouzi[1,2,9], Mona Marzban[2], Nasrin Khazamipour[1,2], Morgan E Roberts [1,2], Negin Farivar[1,2], Igor Moskalev[2], Joey Lo[2], Fariba Ghaidi[2], Irina Nelepcu [2], Alireza Moeen[2], Sarah Truong [2], Robert Dagil [4,5], Swati Choudhary[4,5], Tobias Gustavsson[4,5], Beibei Zhai [1,2], Sabine Heitzender[6], Ali Salanti [4,5], Poul H Sorensen [5,7,8] & Mads Daugaard [1,2,3,5✉]

## Abstract

**Glycosaminoglycans are often deprioritized as targets for synthetic immunotherapy due to the complexity of glyco-epitopes and limited options for obtaining specific subtype binding. Solid tumors express proteoglycans that are modified with oncofetal chondroitin sulfate (CS), a modification normally restricted to the placenta. Here, we report the design and functionality of transient chimeric antigen receptor (CAR) T cells with selectivity to oncofetal CS. Following expression in T cells, the CAR could be "armed" with recombinant VAR2CSA lectins (rVAR2) to target tumor cells expressing oncofetal CS. While unarmed CAR T cells remained inactive in the presence of target cells, VAR2-armed CAR T cells displayed robust activation and the ability to eliminate diverse tumor cell types in vitro. Cytotoxicity of the CAR T cells was proportional to the concentration of rVAR2 available to the CAR, offering a potential molecular handle to finetune CAR T cell activity. In vivo, armed CAR T cells rapidly targeted bladder tumors and increased the survival of tumor-bearing mice. Thus, our work indicates that cancer-restricted glycosaminoglycans may be exploited as potential targets for CAR T cell therapy.**

**Keywords** CAR T Cells; Immunotherapy; Chondroitin Sulfate; Oncofetal CS; Solid Tumor
**Subject Categories** Cancer; Immunology

## Introduction

Chondroitin sulfate (CS) glycosaminoglycans are abundantly present in the placenta where they support the growth and motility of trophoblast cells (Kramer 2010; Clausen, et al, 2016; Van Sinderen et al, 2013). Motility is an essential feature of villous trophoblasts that allow them to invade into the uterine tissue during placental implantation (Abbas et al, 2020; Pollheimer et al, 2018). *Plasmodium falciparum* malaria parasites express VAR2CSA lectins on the exterior of infected red blood cells, mediating unique binding specificity to placental-type CS (Baruch et al, 1995; Salanti et al, 2003; Salanti et al, 2015). In normal physiology, placental-type CS is exclusively present in the placental syncytium and VAR2CSA[+] malaria-infected erythrocytes therefore only accumulate in the placenta.

Cell growth and motility are features shared between trophoblasts and tumor cells (Salanti et al, 2004; Fried and Duffy 1996; Holtan et al, 2009; Baston-Bust et al, 2010). Perhaps to phenocopy features of the placental compartment, tumor cells re-express placental-type CS as a secondary oncofetal modification to a limited repertoire of proteoglycans (Clausen, et al, 2016; Agerbaek et al, 2018; Seiler et al, 2017; Bang-Christensen et al, 2019; Price et al, 2011; Salanti et al, 2015). Accordingly, recombinant VAR2CSA (rVAR2) can be used to guide therapeutic modalities towards solid tumors when formulated as rVAR2–drug conjugates (Salanti et al, 2015; Seiler et al, 2017; Oo et al, 2021; Khazamipour et al, 2020) or bi-specific immune cell engagers (Skeltved et al, 2023; Nordmaj et al, 2021; Khazamipour et al, 2020).

Oncofetal tumor antigens have emerged as potential attractive targets for chimeric antigen receptor (CAR) Tcell therapy (Heitzeneder et al, 2022; Grigor et al, 2017). CARs are genetically engineered synthetic antigen receptors that are able to activate upon antigen recognition, independent of MHC presentation (Barrett, et al, 2015; Zhang et al, 2014). CAR Tcell therapy has

[1]Department of Urologic Sciences, Faculty of Medicine, University of British Columbia, Vancouver, BC, Canada. [2]Vancouver Prostate Centre, Vancouver Coastal Health Research Institutes, Vancouver, BC, Canada. [3]Molecular Pathology & Cell Imaging Laboratory, Vancouver Coastal Health Research Institutes, Vancouver, BC, Canada. [4]Centre for Translational Medicine and Parasitology at Department for Immunology and Microbiology, Faculty of Health and Medical Sciences, University of Copenhagen, Copenhagen University Hospital, Copenhagen, Denmark. [5]VAR2 Pharmaceuticals ApS, Copenhagen, Denmark. [6]Division of Pediatric Hematology/Oncology/Stem Cell Transplantation and Regenerative Medicine, Stanford Cancer Institute, Stanford University School of Medicine, Stanford, CA, USA. [7]Department of Molecular Oncology, British Columbia Cancer Research Centre, Vancouver, BC, Canada. [8]Department of Pathology and Laboratory Medicine, University of British Columbia, Vancouver, BC, Canada. [9]These authors contributed equally: Htoo Zarni Oo, Nader Al-Nakouzi. ✉E-mail: mads.daugaard@ubc.ca

achieved unprecedented success in treating patients with hematopoietic malignancies, such as acute B-cell lymphoblastic leukemia and B-cell lymphomas (Grigor et al, 2017; Deng et al, 2022; Denlinger, et al, 2022; Zhang et al, 2022). However, only a few studies have shown promise for CAR Tcell therapy in solid tumors, such as GD2-CAR T cells for relapsed and refractory Neuroblastoma tumors (Del Bufalo et al, 2023), as well as B7H3-, GD2-, and IL13RA2-CAR T cells for human Glioma (Majzner et al, 2022; Vitanza et al, 2023; Leland et al, 2024). CAR Tcell therapy remains challenging in solid tumors due to variables such as architectural heterogeneity, acquired antigen down-regulation or antigen-loss, or a lack of specific surface tumor antigens (Grigor et al, 2017; Albelda 2024). Hence, the identification of specific antigens in solid tumors is one of several necessary steps needed for extending the clinical utility of CAR Tcell therapy beyond hematopoietic cancers. Until now, CAR Tcell therapy has primarily focused on targeting protein antigens expressed on the surface of cancer cells. However, due to the complex challenges of CAR Tcell therapy in solid tumors, there is increasing interest in exploring other types of target molecules including carbohydrates, glycolipids, and glycoproteins. For example, targeting the glycosylation component of a protein rather than the protein itself offers potential advantages. First, tumor-specific protein glycoforms can offer increased tumor selectivity and possibly limit off-target effects (Barnieh et al, 2023; Kehler et al, 2022; Rossig et al, 2018). Second, a specific glycosylation moiety or pattern can be present on several different proteoglycans simultaneously across cell populations, including tumor stem cells, which may overcome challenges related to heterogeneity and dormancy (Khazamipour et al, 2020). Lastly, proteins that are not normally glycosylated may be subject to disease-specific glycosylation, thereby increasing the available tumor target reservoir (Salanti et al, 2015; Rossig et al, 2018; Reily et al, 2019; Mereiter et al, 2019). In this study, we utilized recombinant VAR2CSA proteins to produce CAR T cells with specificity to oncofetal CS glycosaminoglycans, broadly expressed across various solid tumor types.

# Results

## Design and validation of a transient CAR with selectivity to oncofetal CS glycosaminoglycans

Oncofetal CS glycosaminoglycans have been described in multiple solid tumor types, including sarcoma, lymphoma, glioma, melanoma, pancreatic cancer, lung cancer, colorectal cancer, breast cancer, prostate cancer, and bladder cancer (Salanti et al, 2015; Clausen, et al, 2016; Clausen, et al, 2016; Agerbaek et al, 2017; Seiler et al, 2017; Agerbaek et al, 2018; Bang-Christensen et al, 2019; Clausen et al, 2020; Nordmaj et al, 2021; Oo et al, 2021; Al-Nakouzi et al, 2022; Skeltved et al, 2023; Khazamipour et al, 2020). The oncofetal CS modification can be specifically targeted by rVAR2 proteins that are currently under investigation as vehicles for therapeutic delivery (Salanti et al, 2015; Seiler et al, 2017; Skeltved et al, 2023; Nordmaj et al, 2021; Khazamipour et al, 2020) and as reagents in liquid biopsy diagnostic applications (Agerbaek et al, 2018; Bang-Christensen et al, 2019; Clausen et al, 2020). Indeed, the rVAR2 proteins are specific for oncofetal CS, and binding to tumor cells can be competed with soluble CS, as shown in UM-UC3

bladder cancer cells (Fig. 1A) and tumor cell lines representing both mesenchymal and epithelial origins (Fig. EV1A) (Salanti et al, 2015; Seiler et al, 2017). In primary tumor specimens, oncofetal CS is found in the tumor stroma and on cell membranes, and its presence generally increases with tumor stage (Salanti et al, 2015; Seiler et al, 2017; Agerbaek et al, 2017). For instance, in bladder cancer patients, oncofetal CS presentation is significantly associated with advanced T stage ($P = 0.0231$) and N stage ($P = 0.0114$). (Fig. EV1B–D). Compared to bladder cancer, neuroblastoma tumors presents with lower amounts of oncofetal CS, yet the advanced-stage tumors (II-IV) contain higher oncofetal CS levels as compared to stage I tumors (Fig. EV1E,F). Interestingly, high levels of oncofetal CS were associated with poor survival of neuroblastoma patients (Fig. EV1G), a trend also observed in other cancer types (Seiler et al, 2017; Oo et al, 2021).

With oncofetal CS being a broadly expressed target across solid tumor models, we decided to explore this glycosaminoglycan as a target for CAR Tcell therapy. We first tested the levels of oncofetal CS on naive and activated human T cells using rVAR2 as the detection reagent to assess the risk of potential CAR-induced Tcell self-elimination. Activated and naive T cells expressed minimal-to-undetectable levels of oncofetal CS and the expression was ~20× lower than that detected in UM-UC-3 cells (Fig. 1A).

We next designed a CAR construct that utilized the specificity of rVAR2 for oncofetal CS targeting. The design allowed the CAR to be armed with rVAR2 after expression in the Tcell membrane. This un-conventional design was deployed to alleviate inherent problems with obtaining correct folding of the VAR2-CAR fusion protein in human T cells. The CAR was comprised of the intracellular activation, co-stimulatory, and transmembrane domains CD3zeta, 41BB, and CD28, fused in-frame with a sequence encoding the split-protein intein domain, SpyCatcher (SpyC), derived from *Streptococcus pyogenes* fibronectin-binding protein FbaB (Fig. 1B) (Zakeri et al, 2012). For context, when the SpyCatcher domain comes into contact with the other half of the split-intein, the so-called SpyTag, they spontaneously form a covalent isopeptide bond that irreversibly links the split-intein components (Zakeri et al, 2012; Li et al, 2014). Accordingly, we anticipated this design to allow for the expression of a dormant [SpyC]-CAR in T cells that could then be subsequently armed with rVAR2 genetically fused to a SpyTag, VAR2-[SpyT]. We next transduced human T cells with the chimeric [SpyC]-CAR via lentiviral gene transfer (Fig. 1C). The [SpyC]-CAR sequence includes a Flag-tag that enables detection of CAR expression in T cells. Indeed, [SpyC]-CAR transduction of human T cells was ~90% efficient and was expressed by both CD4+ and CD8+ T-cell populations (Fig. 1D), with <10% of the population staining negative for Flag (Fig. 1E).

We next attempted to arm the [SpyC]-CAR T cells with the recombinant VAR2-[SpyT] warhead (Fig. 1F). The VAR2-[SpyT] recombinant protein contains a V5 tag that enables specific detection of armed CARs via assessment of V5 and Flag double-positive T cells. Here, 68.4% of the [SpyC]-CAR T cells could be armed with the VAR2-[SpyT] protein (Fig. 1G), and the spontaneous VAR2-[SpyT][SpyC]-CAR reaction on T-cell membranes saturated at ~150 nM VAR2-[SpyT] (Fig. 1H). In aggregate, these data demonstrate successful generation of [SpyC]-CAR T cells that can be armed with rVAR2-[SpyT] proteins.

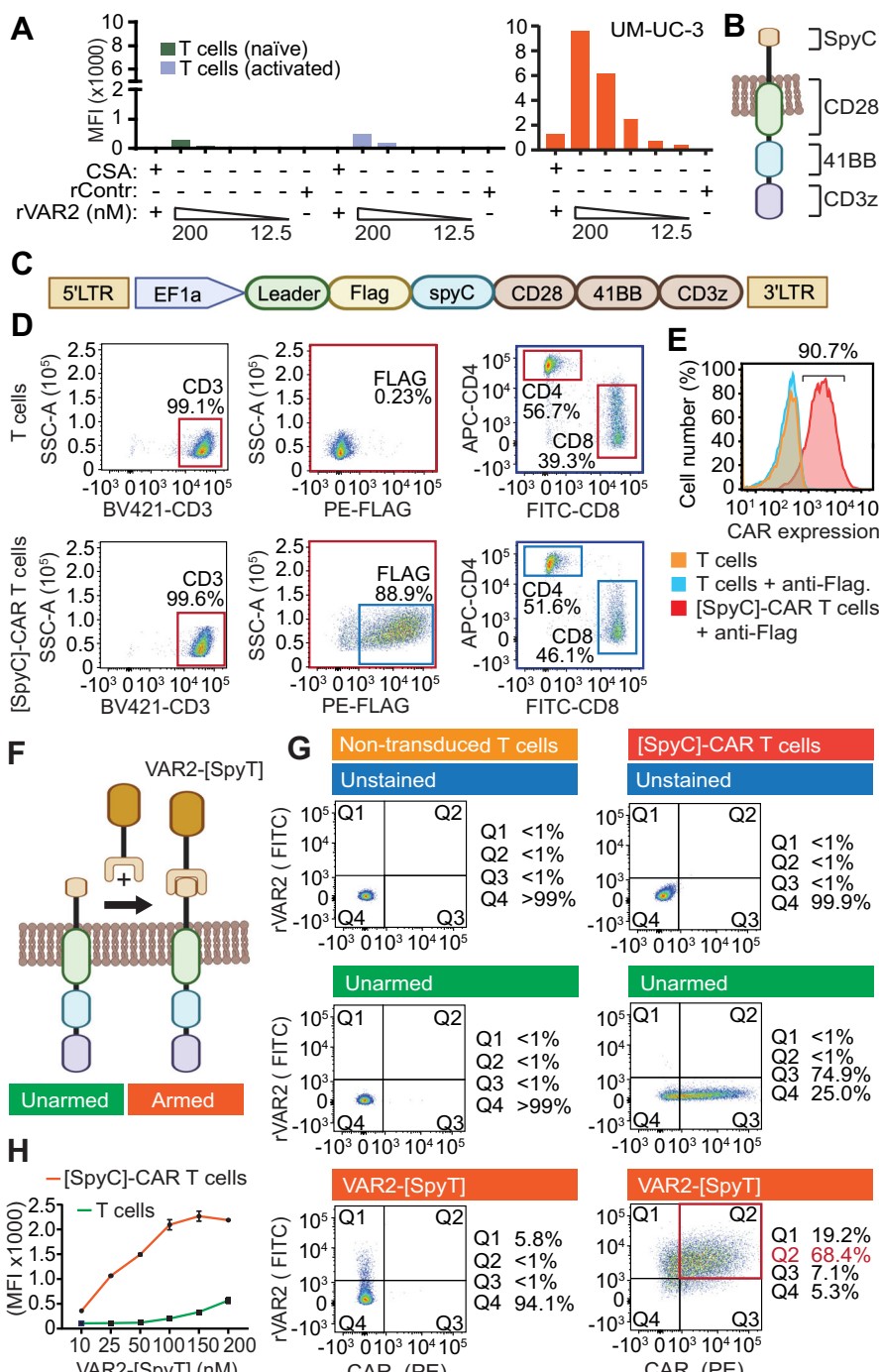

## VAR2-armed CAR T cells activate upon target cell engagement and produce robust cytokine responses

We next investigated whether the armed VAR2-[SpyT][SpyC]-CAR T cells became activated upon engagement with oncofetal CS-positive cancer cells. We used UM-UC-3 muscle-invasive bladder cancer and MG-63 osteosarcoma cells, respectively, as target cell lines for most of the experiments. Upon contact with UM-UC-3 and MG-63 tumor cells, VAR2-[SpyT][SpyC]-CAR T cells, but not unarmed [SpyC]-CAR T cells, upregulated the activation markers CD25 and CD69 (Fig. 2A,B). Similar results were obtained with

LNCaP (prostate cancer) and U2OS (osteosarcoma) cell lines (Fig. EV2A,B). Tcell activation is associated with the induction of cytokines that in turn activate other immune cells, such as macrophages and dendritic cells (Curtsinger and Mescher 2010). Therefore, as a secondary readout for CAR Tcell activation, we examined the expression of key cytokines subsequent to VAR2-[SpyT][SpyC]-CAR Tcell engagement with target tumor cells. Armed VAR2-[SpyT][SpyC]-CAR T cells were able to trigger a robust upregulation of IFNgamma (IFNγ), IL-2, and TNFalpha (TNFα) secretion in all cell lines tested while minimal cytokine levels were detected after exposure to unarmed [SpyC]-CAR T cells

**Figure 1.   Design and validation of a transient CAR with selectivity to oncofetal CS glycosaminoglycans.**

(A) Naive and activated human T cells, and UM-UC3 tumor cells, were incubated with control protein or different concentrations of V5-tagged VAR2 (12–200 nM) protein $+/-$ purified CSA as indicated. Binding of rVAR2 was assessed by flow cytometry using anti-V5-FITC. (B) Illustration of the [SpyC]-CAR containing intracellular CD3zeta, 41BB, and CD28 domains fused with an extracellular split-intein SpyCatcher (SpyC) domain. (C) Schematic diagram of the [SpyC]-CAR DNA construct in a lentiviral plasmid. (D) Isolated human T cells with or without [SpyC]-CAR transduction were analyzed for CD4 and CD8 expression by flow cytometry. (E) [SpyC]-CAR T cells were incubated for 14 days post-transduction and expression of the [SpyC]-CAR (Flag $+$) was assessed by flow cytometry. The result is representative of three individual donors. (F) Schematic of [SpyC]-CAR T-cell arming with the recombinant VAR2-[SpyT] protein. (G) Expression of the [SpyC]-CAR and arming with VAR2-[SpyT] on T cells were assessed by incubating the cells with zero or 200 nM VAR2-[SpyT] in triplicate, staining for the Flag-tag on [SpyC]-CAR and the V5 tag present on (VAR2-[SpyT]), as detected by flow cytometry. [SpyC]-CAR T cells (right panel) were compared to the non-transduced T cells (left panel). The cells are shown in different conditions, unstained, unarmed and incubated with VAR2-[SpyT]. (H) T cells with or without [SpyC]-CAR transduction were incubated with indicated concentrations of VAR2-[SpyT] protein and analyzed for VAR2-[SpyT][SpyC]-CAR assembly by flow cytometry. Error bars shown are mean ± SEM of triplicate wells. All data are representative of three independent experiments. CSA Chondroitin Sulfate A, rContr recombinant control protein, MFI mean fluorescence intensity. Source data are available online for this figure.

or non-transduced T cells (Figs. 2C,D and EV2C,D). Induction of additional cytokines such as IL-4, IL-6, and IL-13 was also detected in all the cell lines tested (Fig. EV3). Combined, these data show that armed VAR2-[SpyT][SpyC]-CAR T cells became activated upon target tumor cell engagement.

## VAR2-[SpyT][SpyC]-CAR Tcell cytotoxicity is target cell type-dependent

We next investigated the relationship between activated VAR2-[SpyT][SpyC]-CAR T cells and target cell cytotoxicity. For this, we added VAR2-[SpyT] armed and unarmed [SpyC]-CAR T cells to UM-UC-3 and MG-63 target cells at a 1:1 effector-to-target cell (E:T) ratio and imaged the co-cultures for 3 days. Armed VAR2-[SpyT][SpyC]-CAR T cells underwent clonal expansion (green cells) upon target cell engagement, while unarmed [SpyC]-CAR T cells remained inactive (Fig. 3A). Moreover, clonal expansion of the VAR2-[SpyT][SpyC]-CAR T cells efficiently eliminated the target tumor cells (red cells), while cells exposed to unarmed [SpyC]-CAR T cells outgrew the culture by day 3. Notably, VAR2-[SpyT][SpyC]-CAR Tcell expansion was detected in MG-63 cultures at day 1, while similar level of expansion was detected at day 2 in UM-UC-3 (Fig. 3A). To further examine potential differences in sensitivity to VAR2-[SpyT][SpyC]-CAR T cells amongst different target cell types, we subjected our diverse cell line panel to VAR2-[SpyT][SpyC]-CAR T cells in an E:T ratio of 1:1 and recorded tumor cell viability over a week. While a decrease in viability of MG-63 cells was observed after 18 h, UM-UC-3 cells needed 36 h of VAR2-[SpyT][SpyC]-CAR Tcell exposure before a reduction in cell viability could be detected (Fig. 3B). LNCaP and U2OS also required 36 h of VAR2-[SpyT][SpyC]-CAR Tcell exposure to exhibit a reduction in cell viability (Fig. EV4A). We next plotted the oncofetal CS expression levels of the different cell types against the expression of key T cell activation markers and cytokines (i.e., CD69$^+$ and IFNγ) following co-culture (Fig. EV4B,C). Combined, these data show that oncofetal CS-positive tumor cells of different lineages can be targeted and eliminated by VAR2-[SpyT][SpyC]-CAR T cells. The data further indicate that the different amounts of oncofetal CS expressed on the target cells all induce sufficient CAR Tcell activation after target cell engagement.

## [SpyC]-CAR T cells can be transiently armed with VAR2-[SpyT] to eliminate tumor cells

The [SpyC]-CAR was designed to allow subsequent and transient arming of [SpyC]-CAR T cells using rVAR2-[SpyT] proteins in a dose-dependent manner (Fig. 4A). To functionally test this transience, we

evaluated [SpyC]-CAR T cells cytotoxicity against UM-UC-3 and MG-63 target cells after exposure to increasing concentrations of VAR2-[SpyT] (0–200 nM) (Fig. 4A). We observed a concentration-dependent decrease in target cell viability over the week after CAR T-cell exposure that reflected the number of VAR2-[SpyT] proteins available for the [SpyC]-CAR T cells during arming (Fig. 4B). A similar trend was observed in LNCaP and U2OS cells (Fig. EV4D). Notably, UM-UC-3 cells seemed slightly less sensitive to VAR2-[SpyT][SpyC]-CAR T cells in the 25–200 nM VAR2-[SpyT] concentration range as compared to MG-63 (Fig. 4B). To further characterize the difference in sensitivity between the target cell lines, we examined VAR2-[SpyT][SpyC]-CAR Tcell cytotoxicity in various effector-to-target cell (E:T) ratios. As expected, higher CAR Tcell effectors relative to target cells resulted in greater cytotoxicity; however, UM-UC-3 cells generally required more effector cells than MG-63 (Fig. 4C), reflecting the lower sensitivity of UM-UC-3 to VAR2-[SpyT][SpyC]-CAR T cells (Fig. 4B,C). Combined, these data show that [SpyC]-CAR T cells can be transiently armed with the VAR2-[SpyT] warhead to confer an E:T ratio-dependent cytotoxicity toward target cells.

## VAR2-[SpyT][SpyC]-CAR T cells inhibit tumor growth in vivo and prolong animal survival

CAR T-cell therapy is challenging in solid tumors due to lack of durable efficacy (Grigor et al, 2017; Albelda 2024). Improvements of CAR T cell efficacy has been pursued by a variety of approaches, including co-targeting immune evasion mechanisms, combining co-stimulatory domains in the CAR construct, using vaccines that target the CAR epitope, and the use of multiple treatments with short-lived CAR T cells (Foster, et al, 2019). However, none of these approaches have resulted in sufficient increases in efficacy to date. We tested the performance of the VAR2-[SpyT][SpyC]-CAR T cells in a solid tumor xenograft mouse model using UM-UC-3 bladder tumor cells. The UM-UC-3 cell line was selected for our in vivo experiments for the following reasons. MG-63 cells disqualified as an in vivo model due to slow growth and low tumor take in mice. UM-UC-3 cells was selected based on a positive correlation between oncofetal CS expression and activation markers CD69+ and IFNγ in vitro (Fig. EV4C). Since the half-life of the VAR2 protein in blood circulation is <10 min, it is challenging to obtain sufficient exposure to the newly proliferated [SpyC]-CAR T cells in mice. Therefore, instead of injecting additional doses of VAR2-[SpyT] protein to arm the [SpyC]-CARs during clonal expansion, we decided to arm the [SpyC]-CAR T cells in vitro, and then inject several doses of armed CARs into mice. Nude mice were inoculated subcutaneously with $1 \times 10^6$ UM-UC-3 cells in their right

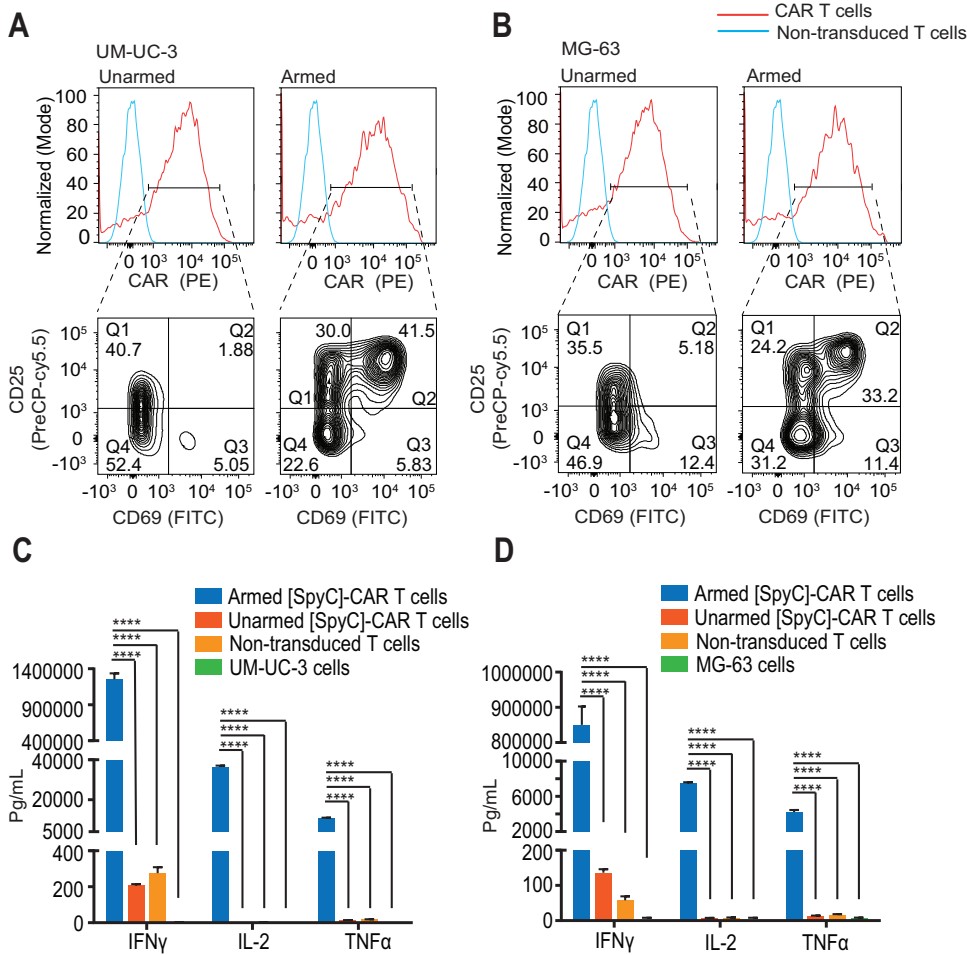

**Figure 2. VAR2-armed CAR T cells activate upon target cell engagement and produce robust cytokine responses.**

(A, B) Armed and unarmed [SpyC]-CAR T cells were incubated in triplicate with (A) UM-UC-3 and (B) MG-63 cells at a 1:1 E:T ratio for 24 h before analysis for expression of Flag, CD69, and CD25 by flow cytometry. The results are representative of 3 independent experiments. (C, D) Armed and unarmed [SpyC]-CAR T cells were incubated with (C) UM-UC-3 and (D) MG-63 cells at a 10:1 E:T ratio in 100 µl media, for 48 h and concentrations of the indicated cytokines in the supernatants were quantified. Results are presented as mean ± SEM of three different wells. The statistical significance was determined using one-way ANOVA, Dunnett's multiple comparison's test. The $P$ value was significant for all comparisons ($P < 0.0001$). E:T effector-to-target cell ratio. Source data are available online for this figure.

flank (day 0). At day 7, the mice were ranked by tumor size and evenly distributed into three groups with nine mice per group. Mice were injected intravenously with PBS (Group 1), unarmed [SpyC]-CAR T cells (Group 2), or armed VAR2-[SpyT][SpyC]-CAR T cells (Group 3) on days 7, 10, 13, 16, and 19 (Fig. 5A). After the first three injections (day 13), the VAR2-[SpyT][SpyC]-CAR T cells (Group 3) started to reduce tumor growth, which became more pronounced over time (Fig. 5B) and was statistically significant as compared to unarmed [SpyC]-CAR T cells (Group 2) (Figs. 5C and EV5). No difference was observed in tumor growth between control groups 1 and 2, indicating no unintentional targeting of tumor cells by unarmed [SpyC]-CAR T cells (Group 2) (Fig. 5C). In the group treated with VAR2-[SpyT][SpyC]-CAR T cells (Group 3), one mouse was tumor-free at the humane endpoint of Group 1 and 2 mice, while the remaining mice had various degrees of treatment benefits as compared to the control groups. This translated into increased overall survival of mice treated with VAR2-[SpyT][SpyC]-CAR T cells (Fig. 5D). In summary,

these data demonstrate a moderate yet significant effect of VAR2-[SpyT][SpyC]-CAR T cells in the treatment of bladder tumors in vivo.

# Discussion

The malaria-derived rVAR2 protein has remarkably high specificity and affinity for oncofetal CS, which is expressed almost exclusively in the placenta and tumors (Salanti et al, 2015). Therefore, targeting oncofetal CS with therapeutic formulations of the VAR2 protein has potentially positive implications for cancer therapy. Oncofetal CS is present in both early and more advanced disease stages, regardless of tumor origin (Seiler et al, 2017) (Oo et al, 2021). The concurrent presence of oncofetal CS across different proteins expressed by tumor cells increases the molecular density of the glycosaminoglycan and the potential sensitivity to oncofetal CS-targeting technologies (Salanti et al, 2015; Clausen, et al, 2016).

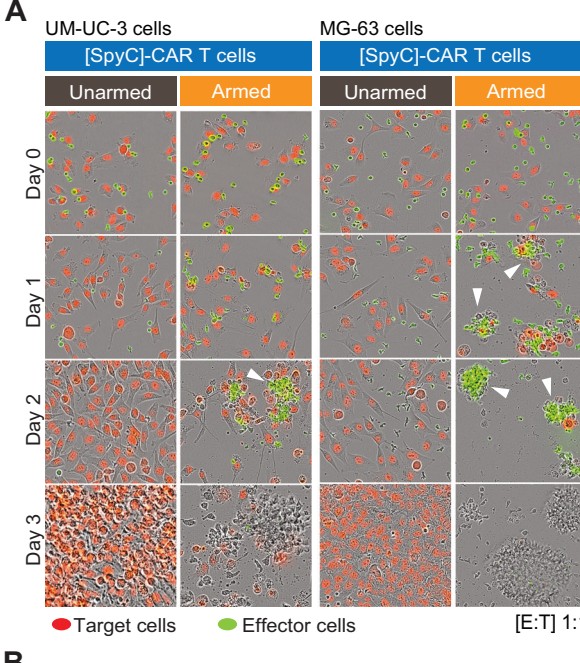

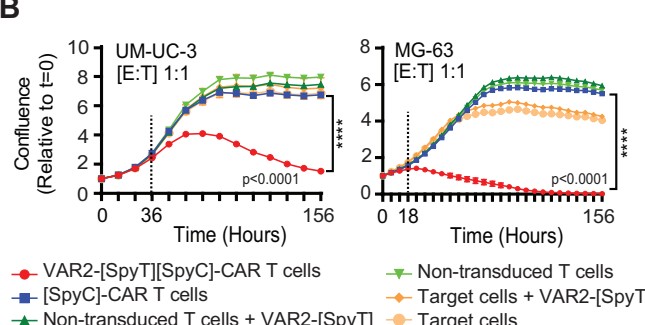

**B**

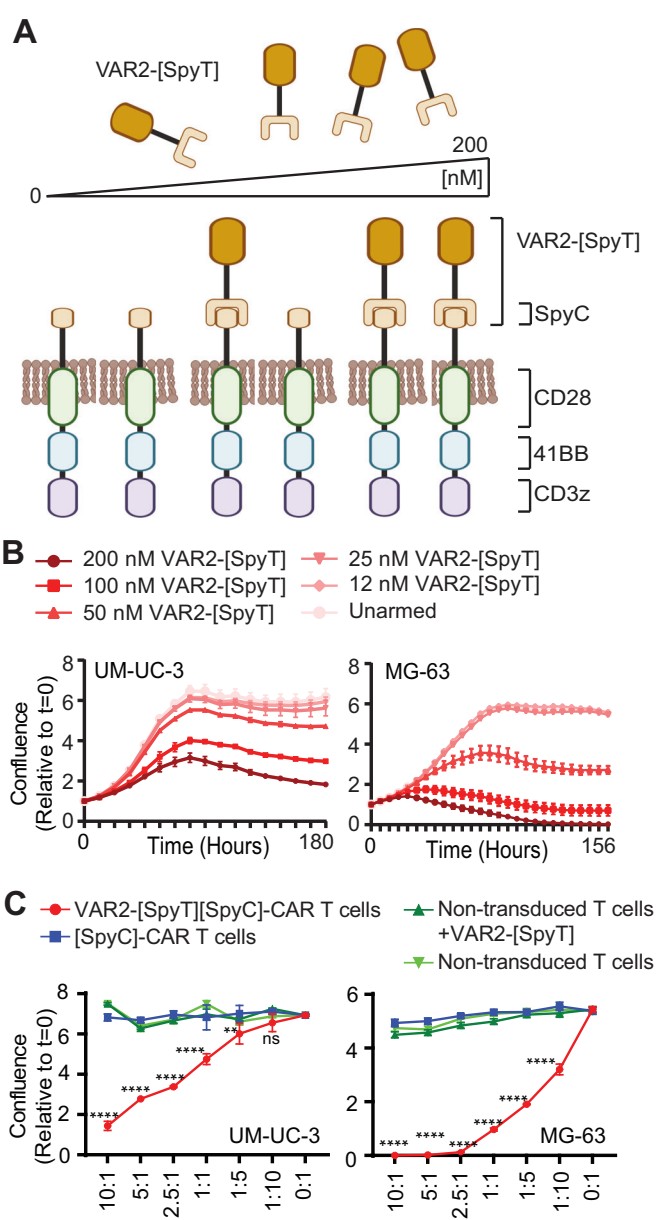

**Figure 3. VAR2-[SpyT][SpyC]-CAR Tcell cytotoxicity is target cell type-dependent and does not directly reflect oncofetal CS content.**

(A) [SpyC]-CAR T cells (green) +/− VAR2-[SpyT] protein were co-cultured with UM-UC-3 and MG-63 target cells (red) at a 1:1 E:T ratio in triplicate and monitored over 3 days. White arrowheads show the CAR Tcell expansion. (B) Red fluorescent expressing UM-UC-3 and MG-63 target cells were co-cultured with indicated formulations of T cells and the confluency of tumor cells monitored over time. Dashed lines indicate time to VAR2-[SpyT][SpyC]-CAR Tcell cytotoxicity. Error bars represent mean ± SEM of triplicate wells. Data from a representative donor of three individual donors are shown. Statistical analyses were performed at the final timepoint using one-way ANOVA with Dunnett's multiple comparisons test. For comparisons of VAR2-[SpyT][SpyC]-CAR Tcell cytotoxicity against all other groups in both cell line, the P value was $P < 0.0001$. E:T effector-to-target cell ratio, MFI mean fluorescence intensity. Source data are available online for this figure.

**Figure 4. [SpyC]-CAR T cells can be transiently armed with VAR2-[SpyT] to eliminate tumor cells.**

(A) Illustration of the transient arming of [SpyC]-CAR T cells with increasing amounts of VAR2-[SpyT] protein. (B) UM-UC-3 and MG-63 target cells (red) were co-cultured with [SpyC]-CAR T cells at a 1:1 E:T ratio in triplicate with the indicated concentrations of VAR2-[SpyT] protein, and assessed for changes in confluence as a readout for tumor cell viability. Error bars represent mean ± SEM of triplicates. (C) UM-UC-3 and MG-63 target cells (red) were co-cultured with [SpyC]-CAR T cells +/− VAR2-[SpyT] in indicated E:T ratios and analyzed as in (B). Data from of three individual donors are shown. Results are presented as mean ± SEM of triplicates. Statistics were calculated using two-way ANOVA and Dunnett's multiple comparisons test. CAR T cells killed UM-UC-3 cells at E:T ratios ranging from 10:1 to 1:1, with a P value of $P < 0.0001$, and at a 1:5 ratio with $P < 0.0095$. No significant difference was observed at the lower concentration (1:10). In MG-63 cells, statistical significance was observed for all E:T ratios from 10:1 to 1:10, with $P < 0.0001$. E:T effector-to-target cell ratio. Source data are available online for this figure.

Preclinical studies in cancer and early-phase clinical trials in malaria have demonstrated the feasibility and safety of rVAR2 as a therapeutic agent. A phase I clinical vaccine trial (PAMVAC) in pregnancy-associated malaria demonstrated low immunogenicity of rVAR2 in humans when administrated without adjuvants, and acceptable safety profiles (Mordmüller et al, 2019). Hence, VAR2CSA may have co-evolved with humans to exhibit minimal immunogenicity as suggested by the persistence of malaria endemics today. Moreover, VAR2 in formulations as drug conjugates or bi-specific immune cell engagers (CD3-fusion

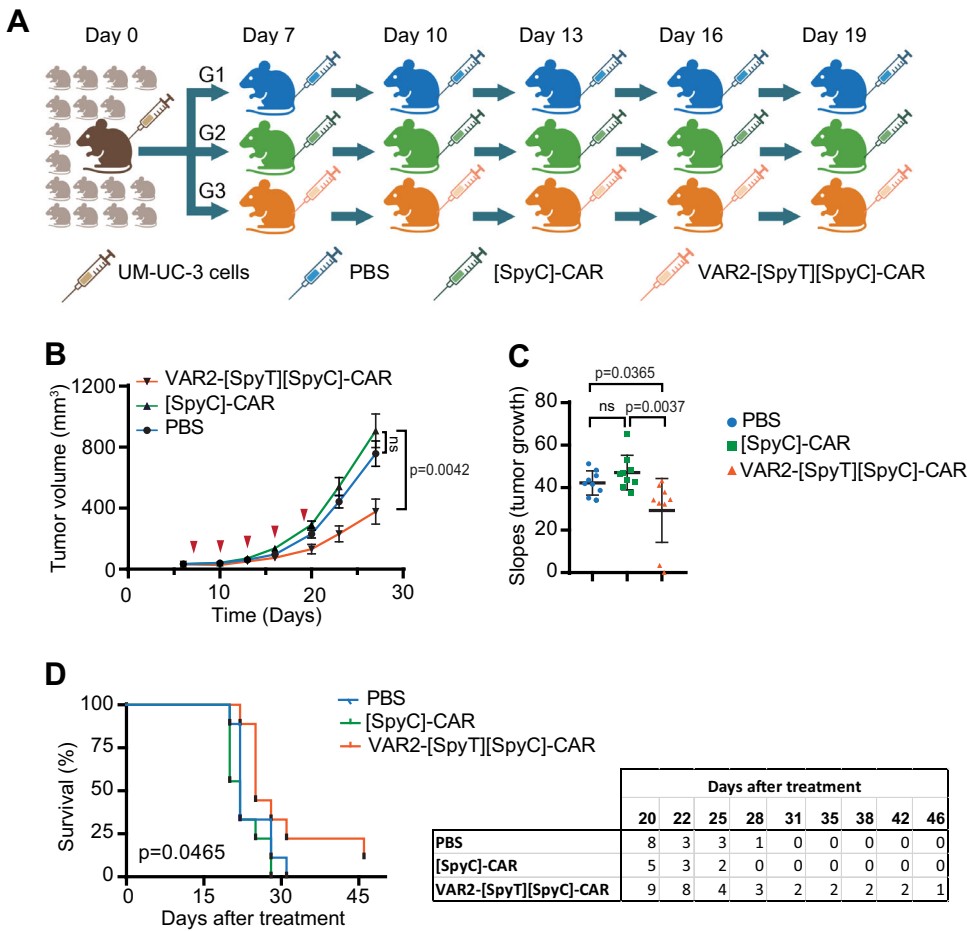

**Figure 5. VAR2-CAR T cells suppress solid tumor growth in vivo and prolong animal survival.**

(A) Schematic outline of the experimental design. Nude mice were inoculated with $1 \times 10^6$ UM-UC-3 tumor cells in their right flank. At day 7, the mice were divided into three groups ($n = 9$). The groups subsequently received five doses of either PBS (G1), 2.5 million [SpyC]-CAR T cells (G2), and 2.5 million VAR2-[SpyT][SpyC]-CAR T cells (G3) by intravenous injection at the indicated time points post-tumor cell inoculation. (B) Tumor growth curves with the mean tumor volumes of mice in each group. Two-way ANOVA followed by a Tukey's multiple comparisons test used to calculate the difference between groups. Data presented as mean ± SEM ($n = 9$ mice). (C) The slope of tumor growth curve for each individual mouse in each group was calculated using linear regression, and the tumor growth rate was compared between treatment groups. The statistical analysis was performed using ANOVA followed by a Tukey's multiple comparison's test. Error bars represent the mean ± standard deviation of the mean (SDM) for tumor slope data from 9 mice. (D) Kaplan–Meier plot (left) indicates morbidity as a proxy for overall survival of mice (numbers of mice) over time (right). Statistical differences between survival curves were analyzed using the log-rank (Mantel–Cox) test. Source data are available online for this figure.

proteins) all exhibit strong efficacies in mouse models with no organ toxicity or immune-related side effects (Salanti et al, 2015; Seiler et al, 2017; Nordmaj et al, 2021).

Since oncofetal CS is widely expressed in solid tumors and the VAR2 protein has been credentialized as a therapeutic vehicle for drug delivery and immune cell engagement, we proposed that VAR2-directed CAR T cells might show efficacy in solid tumors. To test this idea functionally, and to alleviate issues around the safety of current CAR therapies, we designed a CAR construct that could be armed with a recombinant VAR2 protein produced in bacteria after its expression in Tcell plasma membranes. We used the well-defined SpyCatcher-SpyTag intein system derived from *S. pyogenes* (Zakeri et al, 2012) as a molecular glue to link bacterial-produced recombinant VAR2-[SpyT] with [SpyC]-CARs translated and expressed in primary human T cells. [SpyC]-CAR T cells are indolent as they cannot engage target cells and activate. For

activation, the CAR T cells rely on being armed with a SpyTagged binder that can be any molecule (e.g., proteins, peptides, or an scFv) with specificity to a tumor-selective epitope. We used VAR2-[SpyT] protein as the warhead but in principle the [SpyC]-CAR T cells could be armed with any binder of choice. Armed VAR2-[SpyT][SpyC]-CAR T cells were able to fully activate upon engagement with oncofetal CS and eliminated target tumor cells in vitro in a time and dose dependent manner. In a murine xenograft model of bladder cancer, VAR2-[SpyT][SpyC]-CAR T cells were able to curb tumor growth and prolong the survival of the mice. However, with only one complete responder, it is clear that our strategy faces similar obstacles as other CAR T approaches in solid tumors that limits CAR T-cell efficacy, and further work is required to optimize this approach.

A critical consideration for a potential therapeutic application of VAR2-[SpyT][SpyC]-CAR T cells is understanding the

mechanisms underlying incomplete efficacy and potential resistance observed in vivo. Based on our previous work targeting oncofetal CS glycans with VAR2-drug conjugates (VDC) in tumor mouse models, we observed that some tumors re-grew after VDC treatment. Interestingly, relapsed tumors remained positive for oncofetal CS expression, indicating that resistance was not due to the loss of the target but more likely due to sub-optimal dosing. Further analysis of tissue microarrays from various cancer types revealed that some cancer types exhibited small numbers of tumors with low or absent oncofetal CS expression. This suggests that certain tumors can maintain their malignancy without oncofetal CS reprogramming. We hypothesize that these oncofetal CS-low tumors might express a different, currently unknown glycosaminoglycan subtype that can substitute for oncofetal CS, theoretically providing a path for resistance. This hypothesis, although untested, entails that identifying and understanding alternative GAG subtypes might be important for improving the efficacy of oncofetal CS-targeting modalities and overcoming potential resistance.

Upon activation and clonal expansion of the VAR2-[SpyT][SpyC]-CAR T, the daughter cells maintain expression of the [SpyC]-CAR but become unarmed as a result of lack of the warhead during proliferation. The cytotoxicity observed in our study therefore comes from the parental VAR2-[SpyT][SpyC]-CAR T cells after initial target engagement. This implies that CAR T-cell cytotoxicity could potentially be amplified by supplementing excess amounts of VAR2-[SpyT] protein (i.e., re-arming) during the clonal expansion phase in vivo. Consequently, such an approach might offer a safety measure allowing control over CAR T-cell activation based on the availability of VAR2-[SpyT] protein. However, in vivo this strategy is hampered by the fact that the serum half-life of rVAR2 proteins is <10 min, limiting tumor exposure to the point where the concentration of available VAR2-[SpyT] is likely insufficient for saturating newly proliferated [SpyC]-CAR T cells. In vivo re-arming may be possible with formulations that increases rVAR2 plasma half-life. Future work could explore the use of formulations such as PEGylation or albumin-binding sequences to extend the protein's circulation time in vivo. While these strategies might increase potential tumor exposure of the VAR2-[SpyT] protein, they will have to be compatible with the SpyT-SpyC covalent interaction. If this can be achieved, the approach represents a potential avenue for amplifying armed CAR T cells in vivo, thereby improving therapeutic efficacy.

Another consideration for potential clinical utility is the immunogenicity of the CAR component. As with any foreign peptide introduced into the human body, there is an inherent risk of immune responses that could impact the effectiveness or safety of the therapy. However, several factors suggest that counterproductive immunological interference is likely minimal for the VAR2-[SpyT][SpyC]-CAR. Firstly, the VAR2 protein itself is unlikely to present an immunogenic safety issue by itself (Mordmüller et al, 2019). Secondly, the SpyTag component of our fusion protein, comprising only 8 amino acids, has been used extensively in various studies employing the SpyCatcher-SpyTag technology in animals (Lampinen et al, 2023; Hatlem et al, 2019; Sun et al, 2024). To date, there have been no reports of immunogenicity associated with this small peptide sequence. Collectively, we consider the risk of adverse or counterproductive immune reactions to the VAR2-SpyT protein during the course of CAR T-cell treatment low. Nevertheless, while the

VAR2-[SpyT][SpyC]-CAR T cell system offers proof-of-concept for oncofetal CS as a potential target for CAR Tcell therapy, the clinical utility will likely require more traditional CAR Tcell formats using integration of scFv binders against oncofetal CS. Such scFv's have recently been developed and will be tested in CAR T formats moving forward (Vidal-Calvo et al, 2024).

We also observed that the degree of oncofetal CS expression on different tumor cells alone does not appear to determine the activity and cytotoxicity of the VAR2-[SpyT][SpyC]-CAR T cells. This indicates that factors beyond pure target expression might contribute to determining tumor cell sensitivity to CAR T cells. The reasons for this observation can be many, however, a simple explanation could be intrinsic differences amongst the cell lines for sensitivity to Tcell-mediated cell death. We also cannot rule out that epithelial and mesenchymal cell states may have an impact on CAR Tcell efficacy. However, while we often observe that mesenchymal tumor cells present with higher amounts of oncofetal CS (Salanti et al, 2015), experimental testing of lung cancer cells triggered to undergo epithelial-to-mesenchymal or mesenchymal-to-epithelial transitions, did not indicate differences in oncofetal CS content depending on cell states (Agerbaek et al, 2018). Given the current limitations in our ability to control and quantify oncofetal CS modifications at the cellular level, determining a direct cutoff for target epitope presentation facilitating therapeutic activity is challenging. Instead, it might be more feasible to utilize the total number of GalNAc-GlcA disaccharides per cell, as measured by glyco-mass spectrometry, as an indirect yet practical marker for estimating the presence of adequate oncofetal CS. This method, while not directly measuring the number of oncofetal CS chains, provides a quantifiable metric that could serve as a benchmark for sufficient epitopes present to warrant therapeutic activity. Moving forward, enhancing the sensitivity and specificity of bioanalytical techniques to accurately quantify these modifications will be important for refining our approach and ensuring its clinical utility.

In this study, we used female nude mice for our in vivo experiments; however, the potential impact of sex differences on therapeutic outcomes merits consideration. Previous research from our group has demonstrated sex-specific regulatory mechanisms in the biosynthesis of CS, particularly involving hormonal influences. For instance, we have reported that the enzyme CHST11, which is one of several enzymes essential for oncofetal CS synthesis, is regulated by the androgen receptor in prostate cancer, suggesting a direct link between androgen levels and oncofetal CS presentation (Al-Nakouzi et al, 2022). Given this observation, there is evidence to assume that sex could influence the presentation of oncofetal CS in some tumor types and, by extension, the responsiveness to CAR Tcell therapies in such tumors. While we have no evidence of sex-dependent differences in oncofetal CS presentation in bladder cancer, considering sex as a biological variable in future studies is warranted. Understanding how sex hormones influence the tumor microenvironment and the effectiveness of oncofetal CS-targeted CAR Tcell therapy could facilitate more personalized and effective therapeutic utilization.

In summary, we have provided PoC for an oncofetal glycosaminoglycan modification as a target for potential CAR T-cell approaches in solid tumors. Adding cancer-specific glycosaminoglycan modifications to the CAR Tcell target repertoire may provide additional opportunities for immunotherapy in solid tumors.

# Methods

### Reagents and tools table

| Reagent/resource | Reference or source | Identifier or catalog number |
| --- | --- | --- |
| **Experimental models** | | |
| Nude mice | Jackson Laboratory | 002019 |
| Leukopak (*H. sapiens*) | StemCell Technologies | 70500.2 |
| Hek 293T cells (*H. sapiens*) | ATCC | CRL-3216 |
| UM-UC3 cells (*H. sapiens*) | ATCC | CRL-1749 |
| MG-63 cells (*H. sapiens*) | ATCC | CRL-1427 |
| LNCaP cells (*H. sapiens*) | ATCC | CRL-1740 |
| U2OS cells (*H. sapiens*) | ATCC | HTB-96 |
| Neuroblastoma tissue | Patient | |
| Bladder cancer tissue | Patient | |
| **Recombinant DNA** | | |
| pMD2.G envelope | Addgene | 12259 |
| psPAX2 packaging plasmids | Addgene | 12260 |
| Library plasmid | This study | |
| **Antibodies** | | |
| PE-conjugated anti-flag | BioLegend | 637309 |
| PerCP/Cy5.5-conjugated anti CD25 | BioLegend | 302626 |
| FITC-conjugated CD69 | BioLegend | 310904 |
| FITC-conjugated anti-V5 antibody | Invitrogen | R963-25 |
| BV421-conjugated anti-CD3 (Clone SK7) | BD Biosciences | 563798 |
| APC-anti CD4 (Clone OKT4) | StemCell Technologies | 60016AZ.1 |
| FITC anti-human CD8 Antibody | BioLegend | 344703 |
| Mouse monoclonal anti-E-cadherin antibody | BD Biosciences | 610181 |
| Ventana universal secondary antibody | Roche | 760-4205 |
| DISCOVERY DAB Map detection kit | Roche | 760-124 |
| Ventana Ultra ChromoMAP DAB kit | Roche | 760-159 |
| Ventana UltraMap anti-mouse HRP | Roche | 760-4313 |
| V5 Tag Antibody | Invitrogen | 46-0705 |
| **Chemicals, enzymes, and other reagents** | | |
| CellStripper | Corning™ | 23-25-056-CI |
| Incucyte® Nuclight Red Lentivirus | Essen Bioscience | 4475 |
| IncuCyte® CytoLight Rapid Green Reagent | Essen Bioscience | 4705 |
| Zeocin | Gibco | R25001 |
| Puromycin Dihydrochloride | Thermo Fisher Scientific | A1113803 |
| EasySep™ Human T Cell Isolation Kit | StemCell Technologies | 17951 |
| Lymphoprep | StemCell Technologies | 07801 |
| SepMate™ tubes | StemCell Technologies | 85450/85415 |
| human CD3/CD28/CD2 T Cell activator | StemCell Technologies | 10990 |
| IL-2 | StemCell Technologies | 78036.1 |
| Serum-free Opti-MEM | Thermo Fisher Scientific | 11058021 |
| X-tremeGENE HP DNA Transfection Reagent | Sigma Aldrich | 6366236001 |
| ImmunoCult™-XF T Cell medium | StemCell Technologies | 10981 |
| DMEM media | Thermo Fisher Scientific | 10567014 |
| RPMI media | Thermo Fisher Scientific | 11875093 |
| MEM media | Thermo Fisher Scientific | 11095072 |
| DPBS | Sigma Aldrich | D8537-24 |
| MEM Non Essential Amino Acids Solution (100X) | Thermo Fisher Scientific | 11-140-050 |
| MSD V-plex Proinflammatory Panel 1 Human kit | Meso Scale Discovery | K15049D |
| **Software** | | |
| GraphPad Prism 8 | | |
| FlowJo V10.4.2 | | |
| Discovery Workbench software | | |
| **Other** | | |
| IncuCyte S3 | | |
| Flow cytometry | | |

## Cell culture

The cell lines used in this study were maintained in specific culture media tailored to their requirements: T cells were cultured in ImmunoCult XF T-cell Expansion Medium supplemented with 100 U IL-2, UM-UC3 cells were cultured in MEM media supplemented with 1X nonessential amino acids, MG-63 cells were cultured in MEM, LNCaP cells were cultured in RPMI, and U2OS cells were cultured in DMEM. All media were supplemented with 10% fetal bovine serum (FBS). Cultures were maintained at 37 °C incubator with 5% $CO_2$. Regular testing for mycoplasma contamination was performed, and the cell lines were confirmed to be mycoplasma-free prior to use in experiments.

To establish stable cell lines expressing nuclear-restricted mKate2 (a far-red fluorescent protein), the NucLight Lentivirus Reagent (Essen bioscience, Cat. 4476) was utilized for transduction. Cells were seeded in 24-well plates 24 h prior to transduction. The IncuCyte NucLight Lentivirus Reagent was added at a MOI of 3 (= TU/cell), supplemented with 10 μg/mL protamine sulfate. The

plate was incubated at 37 °C, 5% $CO_2$ for 24 h. Media was replaced the next day. After 48 h, cells were treated with zeocin selection marker (Gibco, Cat. R25001) to select for transduced cells.

## Blood samples and T-cell preparation

Leukopak from healthy donors was purchased from StemCell Technologies. T cells were separated by negative selection using EasySep™ Human T Cell Isolation Kit (StemCell, Cat. 17951). The leukopak samples were prepared by adding an equivalent volume of PBS2 and centrifuging at $500 \times g$ for 10 min at room temperature (15–25 °C) and removing the supernatant. The cells were resuspended to a concentration of $5 \times 10^7$ cells/mL in PBS2. T cells were isolated according to the manufacturer. Briefly, the samples were transferred to the polystyrene round-bottom tube, 50 μL/mL EasySep™ Human T Cell Isolation Cocktail and 40 μL/mL of EasySep™ Dextran RapidSpheres™ were added to the samples and incubated for 5 min at RT. The tube was topped up by PBS2, placed in a magnet (StemCell, Cat.18001), and incubated for 3 min. By inverting the magnet, the enriched T-cell suspension was transferred into a clean 14-ml tube. The cells were centrifuged and resuspended in 1 mL ImmunoCult™-XF T Cell Expansion medium and counted to adjust the concentration to $1 \times 10^6$ cells/mL. The cells were activated by incubating with 25 μL/mL of human CD3/CD28/CD2 T Cell activator (StemCell, Cat. 10990) for 3 days at 37 °C with 5% $CO_2$, and then maintained in media containing 100 U/mL IL-2 (StemCell, Cat. 78036.1).

The isolated cells were assessed for T-cell purity and CD4 and CD8 sub-population proportions before and after virus transduction. T cells were stained with BV421-conjugated anti-CD3 (Clone SK7), APC-anti CD4 (Clone OKT4), FITC anti-CD8 (Clone SK1), and PE-FLAG (Clone L5) antibodies for 40 min, washed and subjected to flow cytometry analysis. Unstained and single-color controls were acquired and used for compensation.

## Protein production

Recombinant *P. falciparum* VAR2CSA proteins were produced in *E. coli* SHuffle cells (NEB) and comprised of the minimal CS-binding region with a C-terminal V5 tag and 6x-His tag, and an N-terminal SpyTag. The minimal CS-binding region which consist of the Duffy Binding Ligand-like (DBL) 2× domain with flanking interdomain (ID) regions (subunit ID1-ID2a) has remarkably high specificity and affinity to CS (Salanti et al, 2015). The recombinant control (rContr) protein is made from the non-CS-binding region (DBL4) of VAR2CSA protein with the addition of the C-terminal V5 tag (as in (Salanti et al, 2015)).

## VAR2-binding assay

Tumor cells were cultured in their appropriate media to reach 70–80% confluency. To minimize the risk of damage to proteoglycans, the cells were prepared for binding assay using a non-enzymatic cell dissociation solution (Cellstripper, Cat. CA45000-668). For the preparation of T cells, blood-derived T cells were isolated and divided into two parts. The first part was maintained in culture media without T-cell activators, as the non-activated T cells. The second group was activated in media supplemented with CD2/CD3/CD28 T-cell activator (StemCell, Cat. 10990) (25 μL/mL) and IL-2 (100 U/mL) for a duration of 3 days.

Prior to incubation with VAR2, the cells were washed with PBS containing 2% FBS (PBS2), centrifuged at $350 \times g$ for 5 min, and resuspended to a concentration of $1 \times 10^6$ cells/mL. In total, 100 μl of the cell suspension was added per well in a 96-well plate, followed by centrifugation to pellet the cells. The supernatant was aspirated, and the cells were resuspended in the protein solutions. The protein solutions were prepared by serial dilutions of rVAR2 diluted in PBS2 (12.5–200 nM). A cell sample without rVAR2 was used as background for the antibody signal. 200 nM rContr protein, which is a recombinant non-CS-binding region (DBL4) of the full-length VAR2CSA protein (Salanti et al, 2015), was used as a negative control. Binding specificity was tested by the inclusion of a high concentration (400 μg/ml) of purified CSA (Sigma Cat. 27040) which competes for VAR2 binding. For this competition, 200 nM rVAR2 was pre-incubated with CSA before adding to the cells. The plate was incubated for 30 min at 4 °C on a shaker. After incubation, the cells were washed twice with PBS2 and then stained for 40 min on ice with an anti-V5 antibody conjugated to FITC (Invitrogen, Cat. R963-25). The cells were washed 3 times with 200 μl FACS buffer (PBS containing 2% FBS, 2.5 mM EDTA, and 0.05 mM $NaN_3$) and resuspended in FACS buffer containing DAPI (0.1 μg/mL) for gating out the dead cells. Samples were acquired on a FACS Canto II flow cytometer (BD Biosciences), and the data were analyzed using FlowJo V10.4.2. Unstained and single-color controls were acquired and used for compensation.

## Immunohistochemistry

Freshly cut TMA sections were analyzed for oncofetal CS and E-cadherin expression, using the Ventana Discovery platform. For oncofetal CS staining, sections were incubated in citrate buffer (cell conditioning 2; CC2) at 95 °C for 32 min to retrieve antigenicity and stained with 500 picomolar V5-tagged rVAR2 at room temperature for 12 min, followed by 1:700 mouse monoclonal anti-V5 step, and Ventana UltraMap anti-mouse HRP, then visualize with Ventana Ultra ChromoMAP DAB kit. For an epithelial marker, E-cadherin expression, we used mouse monoclonal anti-E-cadherin (BD Biosciences, catalog# 610181) antibodies. Tissue sections were incubated in Tris EDTA buffer (CC1) at 95 °C for 32 min to retrieve antigenicity, followed by incubation with the E-cadherin antibody (1:200 dilution) at room temperature for 12 h. Bound primary antibodies were incubated with Ventana universal secondary antibody at 37 °C for 32 min and visualized using DISCOVERY DAB Map detection kit.

Interpretation of immunostaining was blinded for clinicopathological parameters and outcome data. Oncofetal CS expression was homogeneous and intensity was determined as score 0–3. Tumor cellular expression score 2 and 3 were considered as oncofetal CS high, while score 0 and 1 as oncofetal CS low. E-cadherin expression was determined as either positive or negative, as the product of staining, and no further data analysis was performed.

## Generation of CAR T cells

Lentivirus particles were generated in HEK293T cells following transfection with the CAR plasmid along with pMD2.G envelope (Addgene, Cat. 12259) and psPAX2 packaging plasmids (Addgene, Cat. 12260), using extreme Xp transfection reagent. In total, $5 \times 10^6$ HEK293T cells were seeded in 10 mL of DMEM media in 10 cm poly-

L-lysine coated culture plates. The next day, the media was replaced with 7 mL of fresh media, 3-4 h before transfection. A total of 25 µg of plasmid DNA, consisting of 10 µg library plasmid, 10 µg envelope plasmid (pMD2.G) and 5 µg packaging plasmid (psPAX2), were diluted in 500 µL serum-free Opti-MEM (Cat.11058021) in a sterile 1.5-mL tube. X-tremeGENE HP DNA Transfection Reagent (Cat. 6366236001) was added in a 3:1 ratio of reagent to DNA, mixed by pipetting and incubated at room temperature for 25 min. The transfection complex was added to cells dropwise while swirling the plate, and the plate was placed in a $CO_2$ incubator. Viral supernatant was collected 48 h and 72 h post transfection and centrifuged at $350 \times g$ for 10 min to remove any cells and debris. The supernatant was filtered through a 0.45-µm Steriflip and used on the same day or stored at $-80\,^{\circ}C$ for future use.

To transduce T cells, the isolated human T cells were centrifuged and resuspended in virus supernatant to a concentration of $1 \times 10^6$ cells/mL in the presence of 10 µg/mL protamine sulfate. 72 h after transduction, 2 µg/mL puromycin was added to the media to select for CAR-expressing T cells.

Two weeks after transduction, the CAR expression levels were quantified by flow cytometry with fluorescent-conjugated Flag antibody. Dead cells were excluded by DAPI staining. The data were analyzed using the FlowJo software to calculate the percentage of the live, single cells with CAR expression.

## VAR2-[SpyT][SpyC]-CAR arming on the Tcell membrane

To assess the optimal quantity of VAR2-[SpyT] required to arm the [SpyC]-CARs on the Tcell membrane, a saturation study was performed. Equal numbers of [SpyC]-CAR T cells ($1 \times 10^5$) were incubated with different concentrations of VAR2-[SpyT] from 0 to 200 nM. The cells were washed three times with PBS2 and stained with FITC-conjugated anti-V5 antibody (at dilution of 1:500) and PE-conjugated anti-Flag antibody (at dilution of 1:100) for 45 min on ice. After staining and washing, cells were resuspended in DAPI-containing FACS buffer for flow cytometry data acquisition. The FlowJo software was utilized for data analysis, initially gating live single cells for CAR expression. Subsequently, the geometric mean fluorescent intensity (MFI) of FITC within the CAR-expressing population was determined to construct the saturation curve. Non-transduced cells were incubated with equivalent amounts of VAR2-[SpyT] and analyzed as the control group. VAR2-[SpyT] arming of the [SpyC]-CAR was also analyzed on flow cytometry by assessing the co-localization of the Flag-tag in CAR construct with the V5 tag on VAR2-[SpyT], following the incubation of the cells with either 200 nM of VAR2-[SpyT] or without VAR2.

## Detection of Tcell activation markers by flow cytometry

The expression of Tcell activation markers, CD25 and CD69, was assessed on effector cells subsequent to encountering their targets. Overall, $1 \times 10^5$ tumor cells were co-cultured with $1 \times 10^5$ [SpyC]-CAR T cells, in the presence (armed CAR) or absence (unarmed CAR) of VAR2-[SpyT]. All samples were prepared in triplicates. The plates were incubated at $37\,^{\circ}C$ overnight, then only the suspension cells were harvested, washed, and stained with PE-conjugated anti-Flag (Clone L5), PerCP/Cy5.5-conjugated anti CD25 (Clone BC96, BioLegend, Cat. 302626), and FITC-conjugated CD69 (Clone FN50, BioLegend, Cat. 310904) antibodies for 30 min

on ice. Unstained and single-color controls were acquired and used for compensation. Following three washes, the samples were resuspended in FACS buffer containing DAPI to exclude the dead cells during subsequent flow cytometry analysis. Unstained and single-color controls were utilized for compensation purpose during analysis using FlowJo software. The cells were first gated for viable single cells that were Flag positive, indicating CAR-expressing cells, and then quantified for upregulation of CD69 and CD25.

## Cytokine analysis

Cytokine production was evaluated using MSD V-plex Proinflammatory Panel 1 Human kit (Meso Scale Discovery, Cat. K15049D) according to the manufacturer's instruction. Effector and target cells were co-cultured at an E:T ratio of 10:1 in 100 µl media in a 96-well plate and incubated for 48 h. The CAR was armed with 200 nM of VAR2-[SpyT]. Post-incubation, media from each well was collected, centrifuged at $350 \times g$ for 5 min at $4\,^{\circ}C$ to remove cells and debris. Supernatants were collected stored in $-80\,^{\circ}C$ freezer, and then evaluated for cytokine levels according to the manufacturer's protocol. Data analyses were performed using Discovery Workbench software.

## In vitro cytotoxicity assay

Red fluorescent tumor cells ($mKate2^+$) were co-cultured with [SpyC]-CAR in triplicate at an E:T ratio of 1:1 in the presence or absence of VAR2-[SpyT]. Similar co-cultures involving non-transduced T cells, as well as the same number of tumor cells without effector cells were used as controls. The plates were placed in the IncuCyte S3 instrument for scanning every 6 h for up to 7 days. At each timepoint, five images per well at ×10 magnification were collected. Total red area (µm²/well) was quantified as a measure of live tumor cells and values were normalized to $t = 0$ measurement. For tracking co-localization of the effector cells with target cells, [SpyC]-CAR T cells were labeled with IncuCyte® CytoLight Rapid Green Reagent. These green-labeled T cells were then co-cultured with red fluorescent tumor cells at a 1:1 ratio in 96-well plates, in the presence or absence of VAR2-[SpyT], and scanned by IncuCyte at ×20 magnification for 3 days.

## In vivo model

Female Nu/Nu mice (8 weeks old), were purchased from Jackson Laboratory and housed in the animal care facility at the Jack Bell Research Centre. The animal study was performed in accordance with protocols approved by the Animal Care Committee at the University of British Columbia (A20-0063). Mice were maintained in ventilated cages (4 mice per cage), with constant humidity (25–47%) and temperature (21–22 $^{\circ}C$), under a 12 h:12 h light:dark cycle, and had ad libitum access to rodent chow diet and drinking water.

Mice were subcutaneously inoculated in their right flanks with $1 \times 10^6$ UM-UC-3 cells suspended in PBS and Matrigel® Matrix solution (1:1). At day 7 post-inoculation, mice were evenly distributed into three groups by tumor size, with nine mice allocated to each group. Investigators were blinded to group allocation. No mice were excluded. The first group received five intravenous (IV) injection of PBS, the second group received five

## The paper explained

### Problem

Chimeric antigen receptor (CAR) Tcell therapy has enjoyed significant success in treating hematopoietic malignancies, but their performance in solid tumor cancers remains challenging due to immune evasion mechanisms, tumor heterogeneity, and the lack of specific surface tumor antigens. This study aims to address these challenges by targeting oncofetal chondroitin sulfate (CS) glycosaminoglycans, uniquely expressed in solid tumors but absent in normal adult tissues.

### Results

We report the development and proof-of-concept of a novel transient CAR Tcell system that can be conditionally armed with recombinant VAR2CSA lectins (VAR2) to target oncofetal CS glycans on tumor cells. VAR2-armed CAR T cells exhibited robust activation and conferred cytotoxicity against various tumor cell types in vitro proportional to the presence of VAR2 doses. In a murine xenograft model of bladder cancer, VAR2-armed CAR T cells inhibited tumor growth and improved survival as compared to control groups.

### Impact

This study introduces oncofetal CS as a potential target for CAR Tcell therapy. The specific expression of oncofetal CS across different solid tumor types may enable the broad applicability of oncofetal CS-targeting CAR T cells. Conditional regulation of CAR Tcell activity through transient availability of the VAR2 lectin serves as a proof-of-concept system for potential CAR Tcell immunotherapies targeting oncofetal CS. Our work adds cancer-specific glycosaminoglycans to the limited repertoire of selective tumor antigens to be explored as targets for CAR Tcell therapy in solid tumor types.

doses of 2.5 million [SpyC]-CAR T cells, and the third group was injected with five doses of 2.5 million VAR2-[SpyT][SpyC]-CAR T cells. The tumor sizes were measured twice per week using a caliper, and tumor volume was calculated by $V = W \times L \times H \times pi/6$ formula, where $W$ = width of tumor, $L$ = length of tumor and H = height of the tumor. The mice were euthanized when reaching tumor size of 1000 mm³. Statistical analysis was performed using GraphPad Prism (GraphPad Software). Two-way ANOVA using Tukey's multiple comparison tests was applied on tumor growth data. Data were presented as mean ± SEM. To quantify the rate of tumor growth over time, we calculated the slope of the tumor growth curve for each individual mouse in each group using linear regression (Figs. 5C and EV5). The statistical significance of the differences between groups was assessed using one-way ANOVA, followed by Tukey's multiple comparison test. In order to compare the survival rates between groups, Kaplan–Meier survival curve was generated, utilizing the morbidity of mice as a surrogate for overall survival across time.

### Statistical analysis

All statistical analyses were performed using GraphPad Prism Software. For comparisons among multiple groups, one-way or two-way ANOVA followed by Tukey's or Dunnett's test were employed to determine statistical significance. Data are presented as mean ± SEM. Survival curves were generated using the Kaplan–Meier method, and statistical differences between survival curves were analyzed using the log-rank (Mantel–Cox) test. A $P$ value of <0.05 was considered statistically significant.

## Data availability

This study includes no data deposited in external repositories.

The source data of this paper are collected in the following database record: biostudies:S-SCDT-10_1038-S44321-024-00153-8.

## Peer review information

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

## Acknowledgements

The authors thank Dr. Crystal Mackall (Stanford, CA) for intellectual input on CAR Tcell methodology, and all members of the Daugaard Lab for helpful discussions. The authors thank the funding support from NIH Prostate Cancer PNW-SPORE (1016339, 223493; 5P50 CA097186-17); Canadian Institutes of Health Research (CIHR) (PJT-153092); St. Baldrick's Foundation/American Association for Cancer Research/Stand Up to Cancer Pediatric Dream Team Translational Research Grant (to PHS and MD; SU2C-AACR-DT-27-17). Stand Up to Cancer (SU2C) is a division of the Entertainment Industry Foundation, and research grants are administered by the American Association for Cancer Research, the scientific partner of SU2C. Nastaran Khazamipour (Nt.K) was supported by the CIHR-Vanier scholarship (#01353-000); UBC Four Year Fellowships (FYF) (#6456); Canadian Urological Association Scholarship Foundation (CUASF-BCC) Research Grant; and the CIHR travel Award-Michael Smith Foreign Study Supplement (#6580) for an internship with Stanford University (CA). AS is supported by NNF Tandem grant (NNF21OC0068192) and NNF Distinguished Innovator grant (NNF22OC0076055). The authors also acknowledge the Terry Fox-funded VPC Molecular Pathology & Cell Imaging Core Facility for their support and services.

## Author contributions

**Nastaran Khazamipour**: Conceptualization; Resources; Data curation; Software; Formal analysis; Supervision; Funding acquisition; Validation; Investigation; Visualization; Methodology; Writing—original draft; Project administration; Writing—review and editing. **Htoo Zarni Oo**: Data curation; Formal analysis; Funding acquisition; Validation; Visualization; Methodology; Writing—review and editing. **Nader AL-Nakouzi**: Formal analysis; Supervision. **Mona Marzban**: Methodology. **Nasrin Khazamipour**: Methodology. **Morgan E Roberts**: Conceptualization; Writing—review and editing. **Negin Farivar**: Methodology. **Igor Moskalev**: Methodology; Writing—review and editing. **Joey Lo**: Methodology. **Fariba Ghaidi**: Methodology. **Irina Nelepcu**: Methodology. **Alireza Moeen**: Investigation; Writing—review and editing. **Sarah Truong**: Writing—original draft; Writing—review and editing. **Robert Dagil**: Investigation; Writing—review and editing. **Swati Choudhary**: Investigation; Writing—review and editing. **Tobias Gustavsson**: Investigation; Methodology. **Beibei Zhai**: Conceptualization; Methodology; Writing—review and editing. **Sabine Heitzender**: Conceptualization; Methodology; Writing—review and editing. **Ali Salanti**: Conceptualization; Funding acquisition; Writing—review and editing. **Poul H Sorensen**: Conceptualization; Funding acquisition; Writing—review and editing. **Mads Daugaard**: Conceptualization; Supervision; Funding acquisition; Validation; Methodology; Writing—original draft; Writing—review and editing.

Source data underlying figure panels in this paper may have individual authorship assigned. Where available, figure panel/source data authorship is listed in the following database record: biostudies:S-SCDT-10_1038-S44321-024-00153-8.

## Disclosure and competing interests statement

MD as the corresponding author certifies that all conflicts of interest, including specific financial interests and relationships and affiliations relevant to the subject matter or materials discussed in the manuscript (e.g., employment/affiliation, grants or funding, consultancies, honoraria, stock ownership or options, expert testimony, royalties, or patents filed, received, or pending), are the following: MD, AS, and PHS are co-founders of, and shareholders in, VAR2 Pharmaceuticals. NAN and TG are consultants for VAR2 Pharmaceuticals. VAR2 Pharmaceuticals is a biotechnology company that specializes in the therapeutic development of the VAR2CSA technology (www.var2pharma.com). The remaining authors declare no competing interests.

# Expanded View Figures

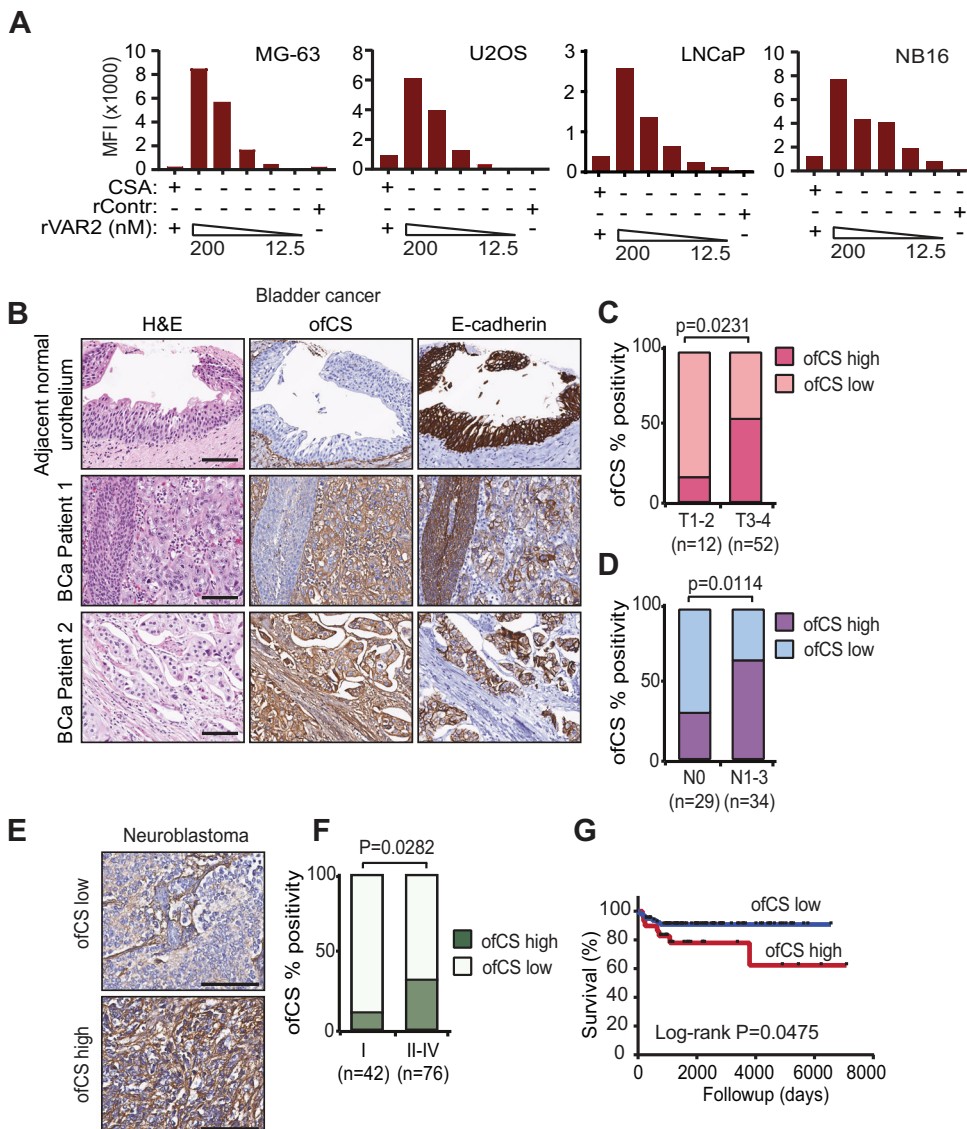

**Figure EV1. Oncofetal CS expression in solid tumor cell lines and bladder cancer tissue.**

(A) MG-63, U2OS, LNCaP, and NB16 tumor cell lines were incubated with indicated concentrations of rContr protein or VAR2 (12–200 nM) $+/-$ purified CSA and analyzed by flow cytometry using anti-V5-FITC. (B) Representative H&E and IHC images of normal adjacent urothelium and bladder cancer tissues from two patients. Matched staining images of E-cadherin, as an epithelial marker, in parallel with oncofetal CS (ofCS) detection by VAR2 and anti-V5. (C) Bar plot of bladder cancer patient tumors ($n = 64$) indicating ofCS expression in relation to T stage. (D) Bar plot of bladder cancer patient tumors ($n = 63$) indicating ofCS expression in relation to N stage. (E) Representative IHC images of neuroblastoma tumors selected for high and low oncofetal CS expression. (F) Percent ofCS-positive neuroblastoma tumors related to tumor stage. (G) Kaplan–Meier plot indicating overall survival of neuroblastoma patients related to oncofetal CS expression. The scale bar represents 100 µm. MIBC: muscle-invasive bladder cancer; oncofetal CS: oncofetal chondroitin sulfate. For statistical analysis in the above panels (C: T stage; D: N stage and F: International Neuroblastoma Staging System (INSS)), two-tailed Fisher's exact test was used. MFI Mean Fluorescence Intensity, CSA chondroitin sulfate A, OfCS Oncofetal Chondroitin Sulfate.

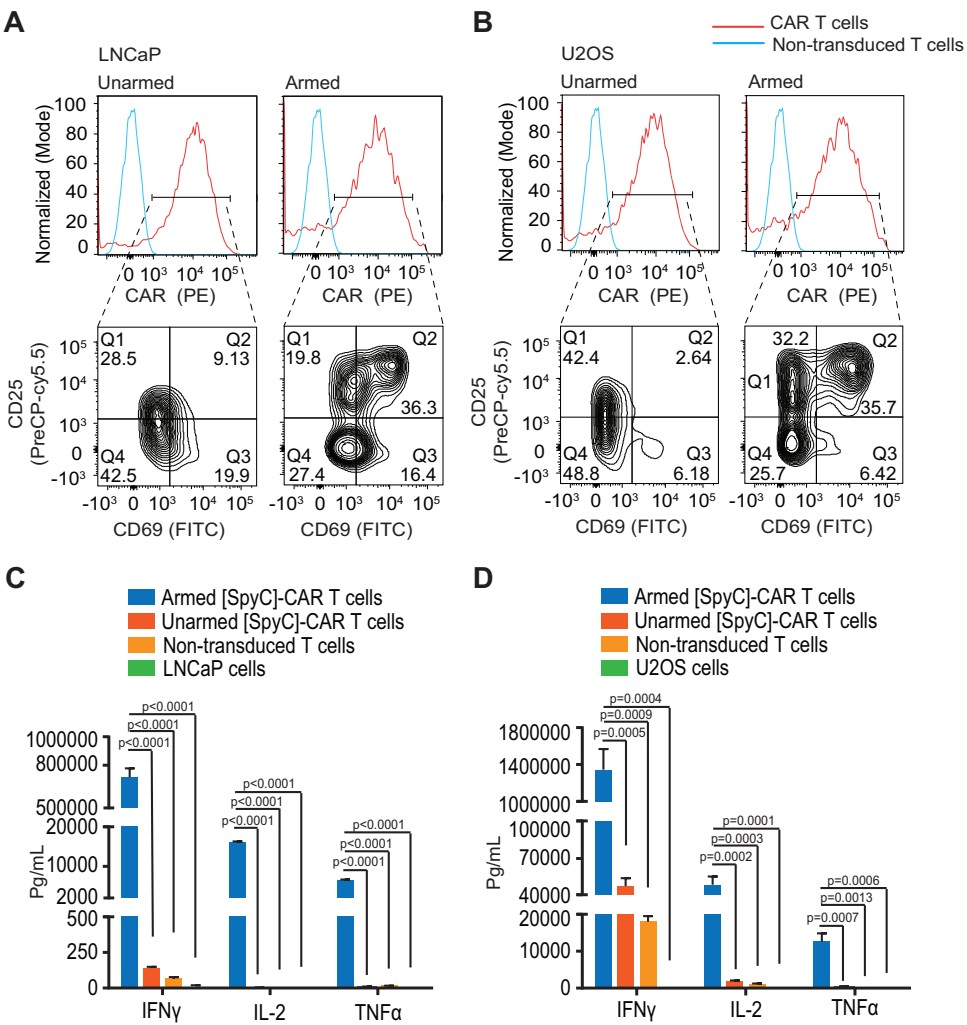

**Figure EV2. Activation of VAR2-[SpyT][SpyC]-CAR T cells upon tumor cell engagement.**

(A, B) Armed and unarmed [SpyC]-CAR T cells were incubated with (A) LNCaP and (B) U2OS cells at a 1:1 E:T ratio for 24 h before analyzed for Flag, CD69, and CD25 expression by flow cytometry. The results are representative of 3 independent experiments. (C, D) Armed and unarmed [SpyC]-CAR T cells were incubated in triplicate, with (C) LNCaP and (D) U2OS cells at a 10:1 E:T ratio in 100 µl media, for 48 h and analyzed for the concentration of indicated cytokines in the culture supernatant. Results are presented as mean ± SEM of three different wells. The statistical significance was determined using one-way ANOVA followed by Dunnett's multiple comparison's test; $P$ values for each comparison are indicated in the figure. E:T effector-to-target cell ratio.

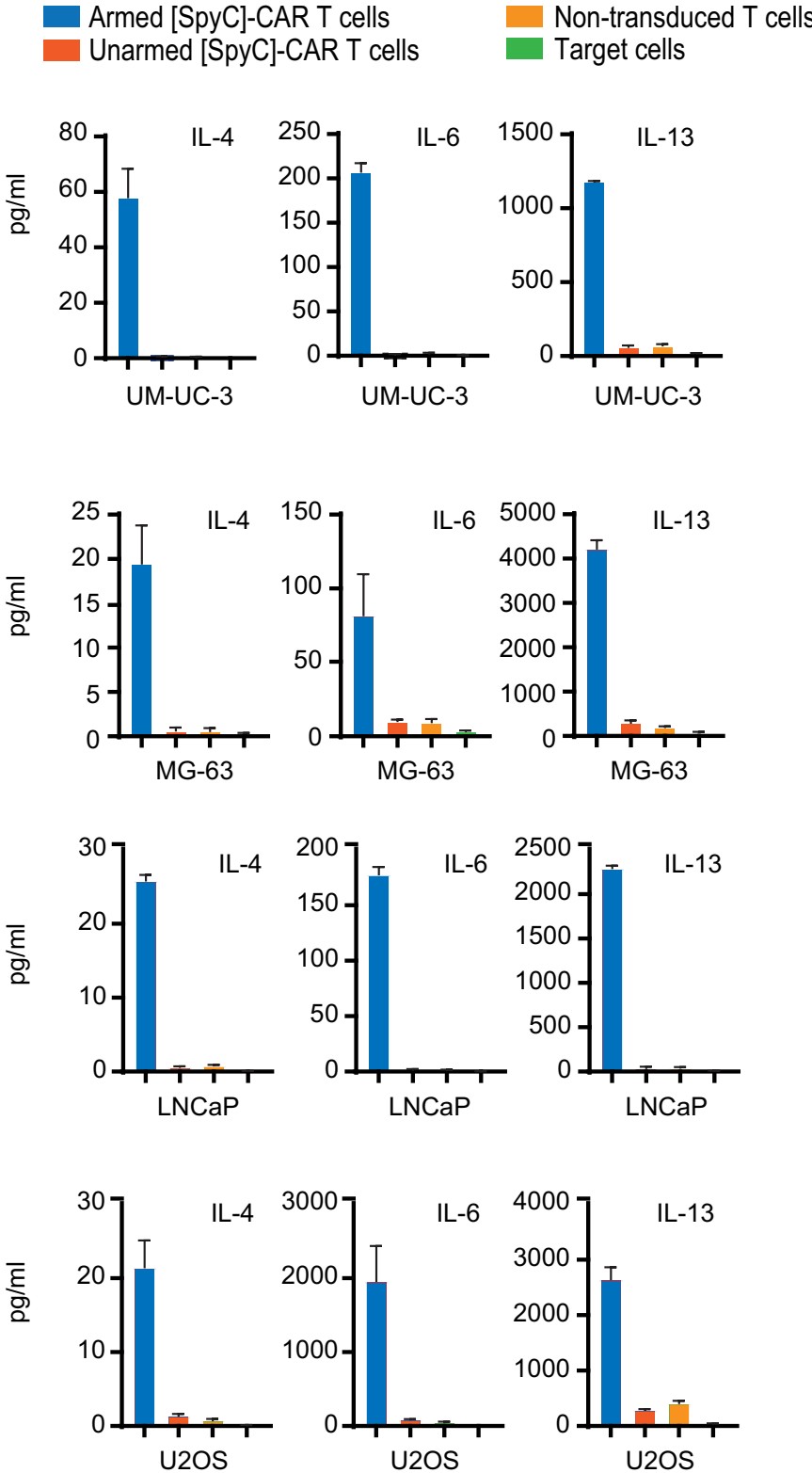

**Figure EV3. Cytokine responses in co-cultures of effector and target cells.**

The concentrations of the indicated cytokines in the culture supernatants were assessed after 48 h of co-culturing between effector T cells formulations and different target cells (i.e., UM-UC3, MG-63, LNCaP, and U2OS), at a 10:1 E:T ratio. Data was analyzed with the Discovery Workbench software. Error bars represent mean ± SEM of three different wells. E:T effector-to-target cell ratio.

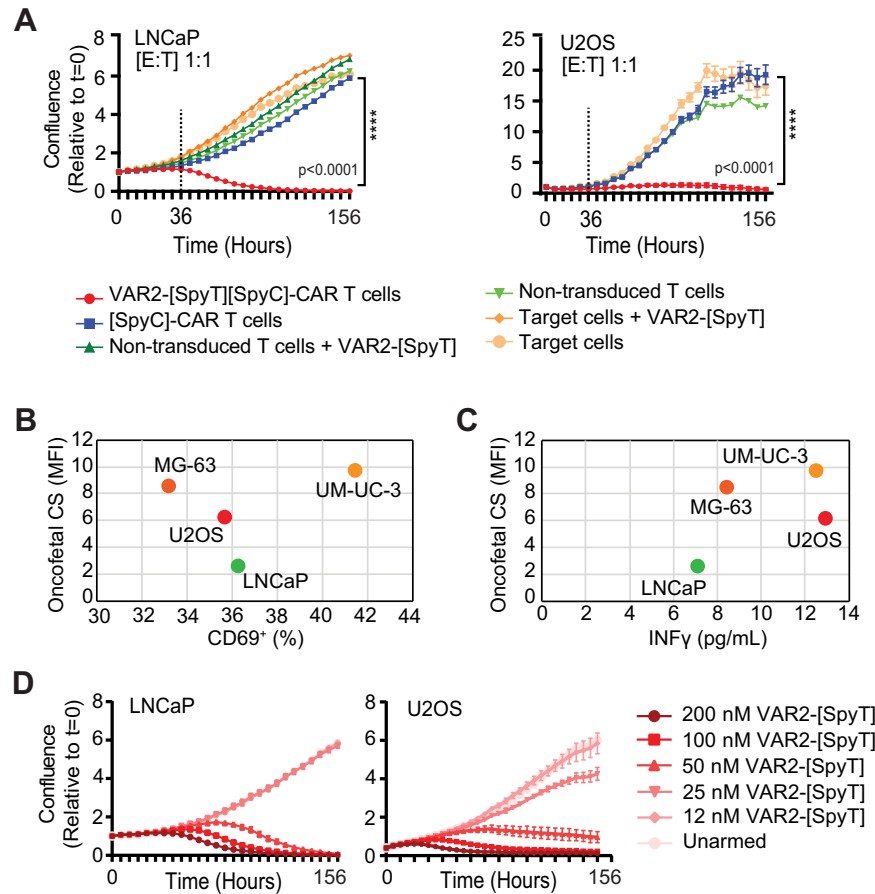

**Figure EV4. Activity of VAR2-[SpyT][SpyC]-CAR T cells after target cell engagement.**

(A) LNCaP and U2OS target cells (red) were co-cultured in triplicate with indicated formulations of T cells and monitored for one week. Dashed lines indicate time to VAR2-[SpyT][SpyC]-CAR T-cell cytotoxicity. Error bars represent mean ± SEM of triplicate wells. Data from a representative donor of three individual donors is shown. Statistical analyses were performed at the final timepoint using one-way ANOVA with Dunnett's multiple comparisons test. The cytotoxicity of VAR2-[SpyT][SpyC]-CAR T cells compared to all other groups in both target cell lines showed a $P$ value of $P < 0.0001$. (B) Percent CD69-positive VAR2-[SpyT][SpyC]-CAR T cells plotted against oncofetal CS expression in indicated target cells. (C) IFNγ production (pg/ml) in co-cultures of VAR2-[SpyT][SpyC]-CAR T cells and indicated target cells plotted against oncofetal CS expression of the target cells. All data was analyzed by GraphPad Prism Software. (D) LNCaP and U2OS target cells (red) were co-cultured with [SpyC]-CAR T cells at a 1:1 E:T ratio with indicated concentrations of VAR2-[SpyT] protein and analyzed for viability using confluence as the readout. Error bars represent mean ± SEM of triplicate wells. E:T effector-to-target cell ratio, MFI Mean Fluorescence Intensity.

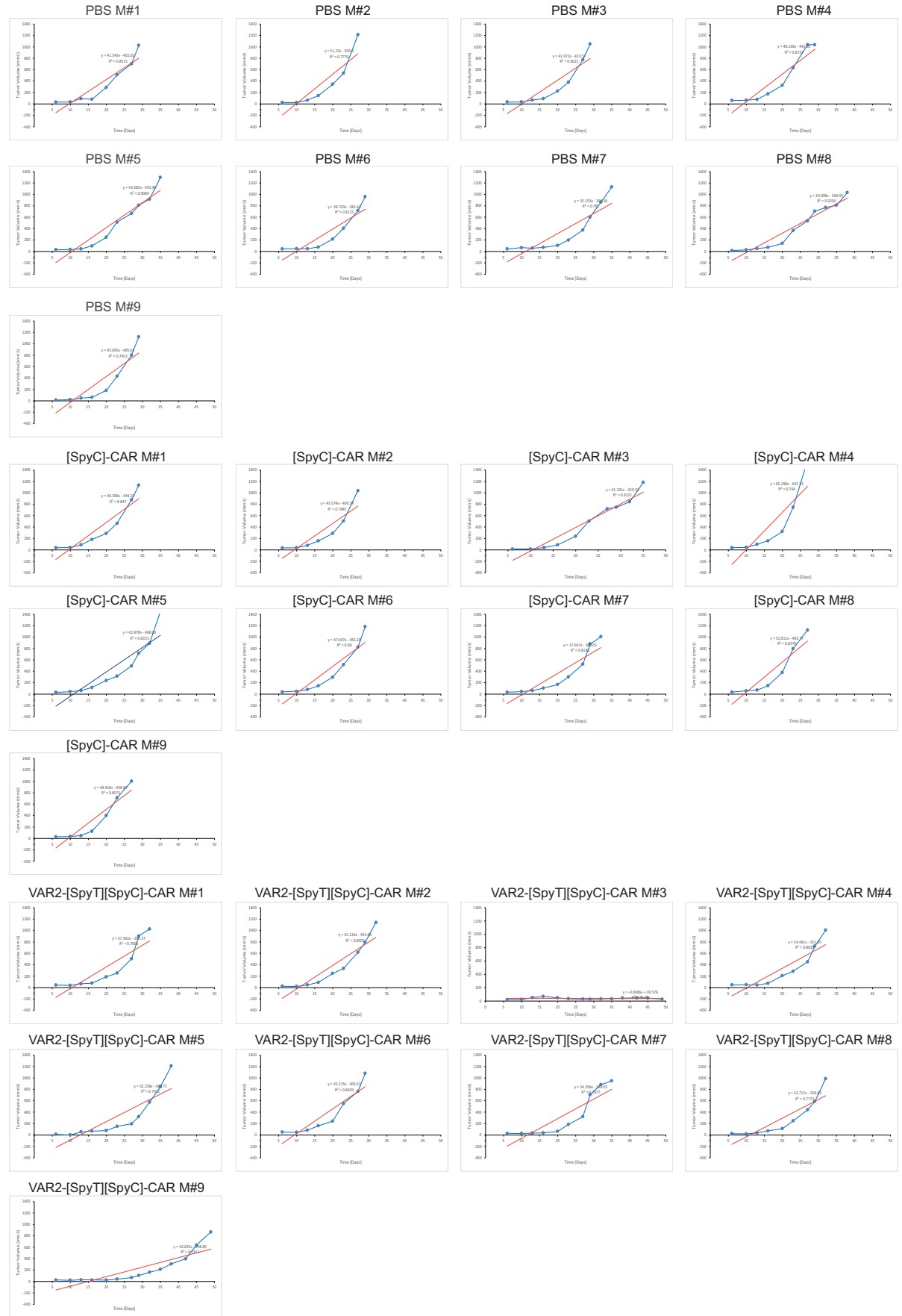

◀ **Figure EV5. Linear regression analysis of tumor growth.**

Individual tumor growth curve (blue line) and the slope of the curve (red line) is shown for each mouse treated with PBS, [SpyC]-CAR T cells or VAR2-[SpyT][SpyC]-CAR T cells.

