## [Peer Review File · EMBO Molecular Medicine]

Transient CAR T cells with specificity to oncofetal glycosaminoglycans in solid tumors

Nastaran Khazamipour, Htoo Zarni Oo, Nader AL NAKOUZI, Mona Marzban, Nasrin Khazamipour, Morgan Roberts, Negin Farivar, Igor Moskalev, Joey Lo, Fariba Ghaidi, Irina Nelepku, Alireza Moeen, Sarah Truong, Robert Dagil, Swati Choudhary, Tobias Gustavsson, Beibei Zhai, Sabine Heitzender, Ali Salanti, Poul Sorensen, and Mads Daugaard

Corresponding author(s): Mads Daugaard (mads.daugaard@ubc.ca) , Mads Daugaard (mads.daugaard@ubc.ca)

Review Timeline:

Submission Date:	31st May 24
Editorial Decision:	28th Jun 24
Revision Received:	16th Jul 24
Editorial Decision:	11th Sep 24
Revision Received:	24th Sep 24
Accepted:	1st Oct 24

Editor: Lise Roth

Transaction Report:

28th Jun 2024

Dear Dr. Daugaard,

Thank you for the submission of your manuscript to EMBO Molecular Medicine. We have now received feedback from the three reviewers who agreed to evaluate your manuscript. As you will see from the reports below, the referees acknowledge the interest of the study and are overall supporting publication of your work pending appropriate revisions.

Addressing the reviewers' concerns in full will be necessary for further considering the manuscript in our journal, and acceptance of the manuscript will entail a second round of review. EMBO Molecular Medicine encourages a single round of revision only and therefore, acceptance or rejection of the manuscript will depend on the completeness of your responses included in the next, final version of the manuscript. For this reason, and to save you from any frustrations in the end, I would strongly advise against returning an incomplete revision.

We are expecting your revised manuscript within three months, if you anticipate any delay, please contact us.

We require:

4) A .docx formatted letter INCLUDING the reviewers' reports and your detailed point-by-point responses to their comments. As part of the EMBO Press transparent editorial process, the point-by-point response is part of the Review Process File (RPF), which will be published alongside your paper.

5) A complete author checklist, which you can download from our author guidelines (<https://www.embopress.org/page/journal/17574684/authorguide#submissionofrevisions>). Please insert information in the checklist that is also reflected in the manuscript. The completed author checklist will also be part of the RPF.

6) All Materials and Methods need to be described in the main text using our 'Structured Methods' format, which is required for all research articles. According to this format, the Methods section includes a Reagents and Tools Table (listing key reagents, experimental models, software and relevant equipment and including their sources and relevant identifiers) followed by a Methods and Protocols section describing the methods using a step-by-step protocol format. The aim is to facilitate adoption of the methodologies across labs. More information on how to adhere to this format as well as a downloadable template (.docx) for the Reagents and Tools Table can be found in our author guidelines:
<https://www.embopress.org/page/journal/17574684/authorguide#structuredmethods>

7) Please note that all corresponding authors are required to supply an ORCID ID for their name upon submission of a revised manuscript.

8) It is mandatory to include a 'Data Availability' section after the Materials and Methods. Before submitting your revision, primary datasets produced in this study need to be deposited in an appropriate public database, and the accession numbers and database listed under 'Data Availability'. Please remember to provide a reviewer password if the datasets are not yet public (see <https://www.embopress.org/page/journal/17574684/authorguide#dataavailability>).

9) For data quantification: please specify the name of the statistical test used to generate error bars and P values, the number

(n) of independent experiments (specify technical or biological replicates) underlying each data point and the test used to calculate p-values in each figure legend. The figure legends should contain a basic description of n, P and the test applied. Graphs must include a description of the bars and the error bars (s.d., s.e.m.). Please provide exact p values.

10) Our journal encourages inclusion of *data citations in the reference list* to directly cite datasets that were re-used and obtained from public databases. Data citations in the article text are distinct from normal bibliographical citations and should directly link to the database records from which the data can be accessed. In the main text, data citations are formatted as follows: "Data ref: Smith et al, 2001" or "Data ref: NCBI Sequence Read Archive PRJNA342805, 2017". In the Reference list, data citations must be labeled with "[DATASET]". A data reference must provide the database name, accession number/identifiers and a resolvable link to the landing page from which the data can be accessed at the end of the reference. Further instructions are available at .

11) We replaced Supplementary Information with Expanded View (EV) Figures and Tables that are collapsible/expandable online. A maximum of 5 EV Figures can be typeset. EV Figures should be cited as 'Figure EV1, Figure EV2' etc... in the text and their respective legends should be included in the main text after the legends of regular figures.

12) The paper explained: EMBO Molecular Medicine articles are accompanied by a summary of the articles to emphasize the major findings in the paper and their medical implications for the non-specialist reader. Please provide a draft summary of your article highlighting

13) For more information: There is space at the end of each article to list relevant web links for further consultation by our readers. Could you identify some relevant ones and provide such information as well? Some examples are patient associations, relevant databases, OMIM/proteins/genes links, author's websites, etc...

14) Author contributions: CRediT has replaced the traditional author contributions section because it offers a systematic machine readable author contributions format that allows for more effective research assessment. Please remove the Authors Contributions from the manuscript and use the free text boxes beneath each contributing author's name in our system to add specific details on the author's contribution. More information is available in our guide to authors.

15) Disclosure statement and competing interests: We updated our journal's competing interests policy in January 2022 and request authors to consider both actual and perceived competing interests. Please review the policy <https://www.embopress.org/competing-interests> and update your competing interests if necessary.

16) Every published paper now includes a 'Synopsis' to further enhance discoverability. Synopses are displayed on the journal webpage and are freely accessible to all readers. They include a short stand first (maximum of 300 characters, including space) as well as 2-5 one-sentences bullet points that summarizes the paper. Please write the bullet points to summarize the key NEW findings. They should be designed to be complementary to the abstract - i.e. not repeat the same text. We encourage inclusion of key acronyms and quantitative information (maximum of 30 words / bullet point). Please use the passive voice. Please attach these in a separate file or send them by email, we will incorporate them accordingly.

17) As part of the EMBO Publications transparent editorial process initiative (see our Editorial at <http://embomolmed.embopress.org/content/2/9/329>), EMBO Molecular Medicine will publish online a Review Process File (RPF) to accompany accepted manuscripts.

In the event of accepted acceptance, this file will be published in conjunction with your paper and will include the anonymous referee reports, your point-by-point response and all pertinent correspondence relating to the manuscript. Let us know whether you agree with the publication of the RPF and as here, if you want to remove or not any figures from it prior to publication. Please note that the Authors checklist will be published at the end of the RPF.

EMBO Molecular Medicine has a "scooping protection" policy, whereby similar findings that are published by others during

review or revision are not a criterion for rejection. Should you decide to submit a revised version, I do ask that you get in touch after three months if you have not completed it, to update us on the status.

I look forward to receiving your revised manuscript.

Yours sincerely,

Lise Roth

**** Reviewer's comments ****

Referee #1 (Remarks for Author):

For this work, the author's provided a proof-of-concept for the use of VAR2-armed CAR-T cells against oncofetal chondroitin sulfate (ofCS) expressing cancer cells. As such, this study is clear and of major interest for the topic of immunotherapy in solid cancers. Overall, the research field of this manuscript is interesting and pertinent. Still, there are some issues that must be clarified.

Major concerns:

1. It is unclear why the authors did not report the oncofetal CS expression also in UM-UC3 cell line, since it is the cell line used as a study model in the work presented (Figure S1A).
2. Also, it is unclear why the authors choose only UM-UC-3 cell line for the in vivo experiment instead of MG-63, the most sensitive cell line to the CAR-T cell therapy, in vitro.
3. The reviewer recommends that authors also include experiments using cell lines that do not express the target antigen (ofCS), for instance, by employing cell transfection techniques. This will help to test the specificity of the generated VAR2-armed CAR-T cells. Additionally, using a non-neoplastic cell line to evaluate off-target effects would provide valuable insights and enhance the robustness of the study. These additions could significantly strengthen current findings.
4. In the in vivo experiments why the authors did not included the conditions of non-transduced T-cells, as for the in vitro techniques, for a baseline comparison?
5. Since the "degree of oncofetal CS expression on different tumor cells alone does not appear to determine the activity and cytotoxicity of the VAR2-[SpyT][SpyC]-CAR T cells", in a real-world scenario, how can we precisely estimate the most effective dose of VAR2-[SpyT] CAR-T cells and the most appropriated E:T ratio, according to the tumor type? How do the authors propose to calculate this cutoff?
6. The reviewer acknowledges a limitation of the study, since the authors only used female nude mice. Would the authors expect sex differences?

Minor concerns:

1. On page 4, lines 19 and 21, the authors use the term 'cancer/tumor indications.' However, the reviewer suggests that the authors consider using more precise scientific terminology. For example, the reviewer suggests replacing by "cancer types, cancer sites or cancer models". Please, give a global review to the entire manuscript concerning scientific terminology.
2. On page 4, lines 22-23, the authors refer the use of "naïve and activated human T cells (...)". However, in the referenced Figure 1A, the labels are 'un-activated and activated T cells.' The authors should ensure uniform terminology between the main text and the figures.
3. Please ensure that all abbreviations used in figure captions are clearly defined. This includes specifying abbreviations present in the figure itself, such as those on the graphic axis. For example: 'Figure 1A and S1A: MFI (mean fluorescence intensity); CSA (Chondroitin sulfate A); ofCS (oncofetal chondroitin sulfate). Please review all the figures.
4. Please provide more detail in the Figure 1G caption. With this high number of test conditions, it is difficult for the reader to follow.
5. Please, re-phrase the following sentence: "(...) at orders of magnitude higher levels than that detected in unarmed [SpyC]-

CAR T cells or non-transduced T cells (...)" (page 6, line 7). It is quite confusing for the reader.

6. It is unclear why the authors repeated the graphs for INF γ , IL-2, and TNF- α detection in Fig. 2 and supplementary Fig. S3. Additionally, please address the discrepancies in the y-axis scales. For example, Fig. 2C includes two breaks at 400 and 40,000 pg/mL, while the corresponding graph in Figure S3 for the same cytokine has no breaks, giving the appearance of different results by masking the low values.

7. It is unclear what the authors mean with "spatiotemporal relationship" in page 6, line 14. Please, clarify the statement.

8. Please, define the "E:T ratio" the first time you mention (page 6, line 16).

9. In Fig.3A, since armed VAR2-[SpyT][SpyC]-CAR T cells are green the authors must choose another color scheme for the image legend.

10. Please, pay attention that Fig.S4B must be referenced in the main text before Fig.S4C-D. Otherwise, the authors should reorder the figure panel.

11. Please look at the sentence "The same was observed for oncofetal CS target cells expression and expression of key T cell activation markers (i.e., CD25+/CD69+ and IFN)" (page 7, lines 4-5). The results for CD25 expression are not represented in the Fig.S4C-D. Please, clarify this issue.

12. The reviewer recommends that the authors rephrase the sentence "The data further indicate that the different amounts of oncofetal CS expressed on target T cells all induce sufficient CAR T activation (...)" (page 7, lines 7-9) to improve clarity and scientific precision. Additionally, I suggest a comprehensive review of the main manuscript to ensure the English language is clear, concise, and scientifically accurate throughout.

13. In page 7, line 15, the sentence is relative to Fig.4B. Please, confirm.

14. There is a minor gap on the title of Figure S5, with a missing "S".

15. Statistical analysis of the research paper must be stated in a separate subsection within the "Materials and Methods" section.

16. There is missing information in the materials and methods section. The immunohistochemistry for ofCS and e-Cadherin antibodies are not described anywhere, nor is the way in which the positivity cut-off (high vs low) was defined (fig. S1). Please add this information to the main manuscript

Referee #2 (Remarks for Author):

Khazamipour et al. have generated CAR T cells targeting oncofetal chondroitin sulfate (CS), using a malarial protein VAR2CSA that is known to bind specifically to placental CS. The authors do so by means of a "conditional CAR" that has an extracellular SpyCatcher domain fused to standard CAR transmembrane and intracellular signaling domains. These CAR T cells are then exposed to a recombinant protein comprised of the CS-binding region of the VAR2 protein fused to SpyTag, which will form an irreversible covalent bond with SpyCatcher and thus direct the CAR T cells against the target CS. The authors show that the VAR2-SpyTag+SpyCatcher CAR T cell combination have activity against tumor cell lines expressing the oncofetal CS.

The use of a parasitic protein to target CAR T cells to cancer-specific glycoproteins is an interesting concept. However, it is unclear to me as to why the authors chose this indirect CAR design (SpyCatcher CAR+VAR2-SpyTag) as compared to fusing VAR2 directly onto the CAR. This indirect CAR design seems to decrease the potency of this therapy without providing any benefit and severely undermines the utility of this strategy.

Major concerns

1. Page 4 lines 30-31 state "This un-conventional design was deployed to alleviate inherent problems of expressing a functional VAR2-CAR fusion-protein in human T cells". The authors should elaborate as to what these inherent problems are.

2. The terminology "conditional CAR" implies that this strategy is tunable using the VAR2-SpyTag protein as an on-off switch. However, the authors are unable to show that injecting the VAR2-SpyTag protein can arm circulating CAR T cells, or discontinuing protein infusion can halt CAR T cell function. A more correct term would be "transient CAR" since that seems to be the effect achieved - Fig 5 shows that multiple doses of CAR T cell injections are required to mediate only modest reduction in tumor growth and prolongation of survival.

3. Fig 3B-C: The authors note that the VAR2-targeting CART cells showed more rapid cytotoxicity against the MG-63 cell line than UM-UC-3 despite similar levels of target expression; however this observation has no utility unless they can provide a reason for why this is the case.

4. The specifics of the VAR2-SpyTag protein used in these studies should be more clearly defined, similar to the diagrams of the SpyCatcher CAR in Fig1B-C. Methods state that a minimal CS-binding subunit of the VAR2CSA protein was used, but this should be more immediately obvious to the readers.

Referee #3 (Remarks for Author):

Review of the manuscript EMM-2024-19987

Khazamipour and colleagues provide highly relevant data on a novel approach to tackle cancer by targeting oncofetal chondroitin sulfate (CS) with a novel CART construct. CS represents an agnostic cancer target and clinical translation of this strategy is of high potential. The authors report for the first time the design and functionality of a conditional CAR T cell product with selectivity to oncofetal CS. The authors 'armed' CARTs with recombinant VAR2CSA lectins (rVAR2) to specifically target cancer cells in in vitro and cancers in murine models in vivo and consequently increased survival. Several minor points should be addressed before a revised version of this excellent work can be accepted in EMBO Molecular Medicine.

Minor points:

1. The authors should comment on potential immunogenicity of VAR2-SpyT. How does this potentially reduce the applicability of this approach in vivo?
2. The half-life time of VAR2-SpyT is only 10 minutes, can this be prolonged by modification to improve the arming process in vivo?
3. In vivo there is not a 100% efficacy. How does resistance occur or can be explained?

Khazamipour et al.

Point-by-point response to reviewers

Referee #1 (R1):

For this work, the author's provided a proof-of-concept for the use of VAR2-armed CAR-T cells against oncofetal chondroitin sulfate (ofCS) expressing cancer cells. As such, this study is clear and of major interest for the topic of immunotherapy in solid cancers. Overall, the research field of this manuscript is interesting and pertinent. Still, there are some issues that must be clarified.

Major concerns:

R1-1: It is unclear why the authors did not report the oncofetal CS expression also in UM-UC3 cell line, since it is the cell line used as a study model in the work presented (Figure S1A).

Response R1-1: We apologize for the confusion. The oncofetal CS expression on UM-UC3 cells is part of main Figure 1A because this is the workhorse cell line used throughout the manuscript. The additional cell lines in Figure 1SA are included to show the broadness of oncofetal CS across unrelated cell lines of different origin. In the revised manuscript, we have included a clarifying sentence on this topic on **page 5, lines 14-15**.

R1-2: Also, it is unclear why the authors choose only UM-UC-3 cell line for the in vivo experiment instead of MG-63, the most sensitive cell line to the CAR-T cell therapy, in vitro.

Response R1-2: This is a valid question and we should have explained the reasons for this. Firstly, MG63 cells do not grow very well in vivo in our hands (very slow and sporadic tumor take) and we simply could not get enough take in mice where the tumors would grow in a speed compatible with the treatment window of the conditional CAR T cells. Secondly, of the remaining cell lines, UM-UC3 cells had the best correlation between oncofetal CS and CD69+ and INF γ (Figure S4C), which we reasoned would facilitate reasonable in vivo activity. Lastly, our project was partially funded by a bladder cancer grant (CUA grant) to which we had a level of obligation in terms of bladder cancer focus. In the revised manuscript, we have included a clarifying sentence on this consideration on **page 9, lines 17-21**.

R1-3: The reviewer recommends that authors also include experiments using cell lines that do not express the target antigen (ofCS), for instance, by employing cell transfection techniques. This will help to test the specificity of the generated VAR2-armed CAR-T cells. Additionally, using a non-neoplastic cell line to evaluate off-target effects would provide valuable insights and enhance the robustness of the study. These additions could significantly strengthen current findings.

Response R1-3: Thank you for the suggestion. Because of a large degree of redundancy in the CS enzymatic machinery, it is unfortunately not possible for us to completely remove the modification in cell lines. For example, the sulfotransferases required to add the specific C4S sulfations to the CS chain (CHST11,12,13, and likely also 14) can substitute for each other. Knocking down the more fundamental enzymes for CS chain formation/elongation is lethal in our cell lines. We have previously established systems where we can decrease oncofetal CS with ~75% by expressing the enzyme chondroitinase ABC in the cell lines (Nature Comms, 2022, PMID: 35963852), but when testing the CAR T cells in these systems, there is still enough target remaining for the CAR T cells to partially activate against the target cells. Therefore, it is not possible for us to completely remove the oncofetal CS target from our tumor cell lines as they seem to have developed a strong dependence on the glycosaminoglycan. Perhaps for similar reasons, non-neoplastic cell lines that have been immortalized and domesticated for cell culture growth also present the structure, making them unsuitable to be used as negative controls in the context of the CAR T cells. As such, on this occasion, we have to rely on previous target credentialization in primary human tissues using the recombinant VAR2CSA malaria protein (used

to arm the transient CAR), showing that the oncofetal CS modification is present in most solid tumors, with no presentation in normal tissues, except from the placenta (Cancer Cell, 2015, PMID: 26461094). This finding is corroborated by the fact that VAR2CSA-expressing malaria parasites do not sequester to any other organ in the human body but the placenta (JBC, 2012, PMID: 22570492).

R1-4: In the in vivo experiments why the authors did not included the conditions of non-transduced T-cells, as for the in vitro techniques, for a baseline comparison?

Response R1-4: Yes, correct. The condition we deemed less relevant in vivo was to have non-transduced T-cells + the VAR2-[Spy]. The reason being that we have shown on multiple occasions earlier that recombinant VAR2 alone does not have an effect on tumor growth in vivo (e.g., Cancer Cell, 2015, PMID: 26461094; European Urology, 2017, PMID: 28408175). The point we are trying to make in this manuscript is that CAR T cells expressing an unarmed CAR are inactive, but arming the CAR translates into activity. To answer that particular question, we thought the more appropriate controls would be comparing armed with unarmed CAR T cells, while assuring that the unarmed CAR T cells did not impact tumor growth as compared to the PBS control.

R1-5: Since the "degree of oncofetal CS expression on different tumor cells alone does not appear to determine the activity and cytotoxicity of the VAR2-[SpyT][SpyC]-CAR T cells", in a real-world scenario, how can we precisely estimate the most effective dose of VAR2-[SpyT] CAR-T cells and the most appropriated E:T ratio, according to the tumor type? How do the authors propose to calculate this cutoff?

Response R1-5: That is a key question for a potential clinical path. Our data suggest that there is something at play contributing to CAR T cell activity on cells beyond target presentation on the surface. We don't have a clear understanding of that at the moment. As we have no way of titrating down the number of oncofetal CS modifications in a controlled manner, it will be difficult to calculate a minimum required number of modifications (oncofetal CS epitopes) to be used as a cut-off. Instead, it might be more feasible to establish an amount of oncofetal CS modifications that evidently will be enough and set this as the bar for potential therapeutic deployment. However, as we do not have a way to detect number of oncofetal CS chains, but can only determine the total number of GalNAc-GlcA disaccharides in glyco-mass spectrometry, the number would have to be expressed as 'number of disaccharides/cell'. In the revised manuscript, we have included a discussion of this important topic on **page 13, lines 8-17**.

R1-6: The reviewer acknowledges a limitation of the study, since the authors only used female nude mice. Would the authors expect sex differences?

Response R1-6: This is a highly relevant question. It is very likely that there will be differences based on sex. We have previously published that the C4S sulfortransferase CHST11 (that is one of many enzymes required for making the oncofetal CS structure) is regulated by the androgen receptor in prostate cancer (Nature Comms, 2022, PMID: 35963852), so in prostate cancer there is a hormonal factor that impacts oncofetal CS presentation. We also have unpublished data in breast cancer that shows estrogen receptor-dependent regulation of a different enzyme in the CS synthesis pathway, that is likely to affect oncofetal CS presentation. So, in summary, we have reasons to believe that sex may impact presentation of oncofetal CS in at least some tumor types. Although we do not have any direct evidence for sex-dependent oncofetal CS presentation in bladder cancer cells (e.g., UM-UC3), we have chosen to include a discussion point on this topic in the revised manuscript on **page 13-14, lines 20-31, 1-2**.

Minor concerns:

R1-7: On page 4, lines 19 and 21, the authors use the term 'cancer/tumor indications.' However, the reviewer suggests that the authors consider using more precise scientific terminology. For example, the reviewer suggests replacing by "cancer types, cancer sites or cancer models". Please, give a global review to the entire manuscript concerning scientific terminology.

Response R1-7: Point taken. Thank you. In the revised manuscript, we have corrected occasionally vague descriptions with more precise terminology.

R1-8: On page 4, lines 22-23, the authors refer the use of "naïve and activated human T cells (...)". However, in the referenced Figure 1A, the labels are 'un-activated and activated T cells.' The authors should ensure uniform terminology between the main text and the figures.

Response R1-8: Resolved.

R1-9: Please ensure that all abbreviations used in figure captions are clearly defined. This includes specifying abbreviations present in the figure itself, such as those on the graphic axis. For example: 'Figure 1A and S1A: MFI (mean fluorescence intensity); CSA (Chondroitin sulfate A); ofCS (oncofetal chondroitin sulfate). Please review all the figures.

Response R1-9: Resolved.

R1-10: Please provide more detail in the Figure 1G caption. With this high number of test conditions, it is difficult for the reader to follow.

Response R1-10: Additional information now included in the Figure 1G caption.

R1-11: Please, re-phrase the following sentence: "(...) at orders of magnitude higher levels than that detected in unarmed [SpyC]-CAR T cells or non-transduced T cells (...)" (page 6, line 7). It is quite confusing for the reader.

Response R1-11: Thank you for the suggestion. The sentence now reads *"Armed VAR2-[SpyT][SpyC]-CAR T cells were able to trigger a robust upregulation of IFN γ (IFN γ), IL-2, and TNF α (TNF α) secretion in all cell lines tested while minimal cytokine levels were detected after exposure to unarmed [SpyC]-CAR T cells or non-transduced T cells".*

R1-12: It is unclear why the authors repeated the graphs for INF γ , IL-2, and TNF- α detection in Fig. 2 and supplementary Fig. S3. Additionally, please address the discrepancies in the y-axis scales. For example, Fig. 2C includes two breaks at 400 and 40,000 pg/mL, while the corresponding graph in Figure S3 for the same cytokine has no breaks, giving the appearance of different results by masking the low values.

Response R1-12: This was an error on our side. Thank you for picking it up. Figure 2C and Figure S3 have been corrected in the revised manuscript. In Figure S3, we deleted the graphs for INF γ , IL2 and TNF α for all cell lines, as we have them in main Figure 2C and S2C-D. The graph-breaks in main Figures are to visualize the smaller values of the control conditions.

Khazamipour et al.

R1-13: It is unclear what the authors mean with "spatiotemporal relationship" in page 6, line 14. Please, clarify the statement.

Response R1-13: We agree that the word 'spatiotemporal' is confusing and unnecessary. In the revised manuscript, we have omitted the word and the sentence now reads: "*We next investigated the relationship between activated VAR2-[SpyT][SpyC]-CAR T cells and target cell cytotoxicity*".

R1-14: Please, define the "E:T ratio" the first time you mention (page 6, line 16).

Response R1-14: Resolved.

R1-15: In Fig.3A, since armed VAR2-[SpyT][SpyC]-CAR T cells are green the authors must choose another color scheme for the image legend.

Response R1-15: Yes, agreed. New colors in the revised Figure 3A.

R1-16: Please, pay attention that Fig.S4B must be referenced in the main text before Fig.S4C-D. Otherwise, the authors should reorder the figure panel.

Response R1-16: Thank you. Figure S4 have been reorganized accordingly and corrected in the text.

R1-17: Please look at the sentence "The same was observed for oncofetal CS target cells expression and expression of key T cell activation markers (i.e., CD25+/CD69+ and IFN γ)" (page 7, lines 4-5). The results for CD25 expression are not represented in the Fig.S4C-D. Please, clarify this issue.

Response R1-17: This was a mistake on our side. The Figures show CD69 and IFN γ only so we have removed the word 'CD25' and added 'cytokine production' in the main text.

R1-18: The reviewer recommends that the authors rephrase the sentence 'The data further indicate that the different amounts of oncofetal CS expressed on target T cells all induce sufficient CAR T activation (...)' (page 7, lines 7-9) to improve clarity and scientific precision. Additionally, I suggest a comprehensive review of the main manuscript to ensure the English language is clear, concise, and scientifically accurate throughout.

Response R1-18: We have reviewed and corrected this particular issue, as well as a number of additional sentences in the revised manuscript.

R1-19: In page 7, line 15, the sentence is relative to Fig.4B. Please, confirm.

Response R1-19: That is correct.

R1-20: There is a minor gap on the title of Figure S5, with a missing "S".

Response R1-20: Resolved.

R1-21: Statistical analysis of the research paper must be stated in a separate subsection within the "Materials and Methods" section.

Response R1-21: A separate sub-section on statistics has been included in the M&M section on page 22, lines 24-30.

R1-22: There is missing information in the materials and methods section. The immunohistochemistry for ofCS and e-Cadherin antibodies are not described anywhere, nor is the way in which the positivity cut-off (high vs low) was defined (fig. S1). Please add this information to the main manuscript.

Response R1-22: Thank you for picking up this error. This information is included in the revised manuscript on page 19, lines 2-18.

Referee #2 (R2):

Khazamipour et al. have generated CAR T cells targeting oncofetal chondroitin sulfate (CS), using a malarial protein VAR2CSA that is known to bind specifically to placental CS. The authors do so by means of a "conditional CAR" that has an extracellular SpyCatcher domain fused to standard CAR transmembrane and intracellular signaling domains. These CAR T cells are then exposed to a recombinant protein comprised of the CS-binding region of the VAR2 protein fused to SpyTag, which will form an irreversible covalent bond with SpyCatcher and thus direct the CAR T cells against the target CS. The authors show that the VAR2-SpyTag+SpyCatcher CAR T cell combination have activity against tumor cell lines expressing the oncofetal CS.

The use of a parasitic protein to target CAR T cells to cancer-specific glycoproteins is an interesting concept. However, it is unclear to me as to why the authors chose this indirect CAR design (SpyCatcher CAR+VAR2-SpyTag) as compared to fusing VAR2 directly onto the CAR. This indirect CAR design seems to decrease the potency of this therapy without providing any benefit and severely undermines the utility of this strategy.

Major concerns:

R2-1: Page 4 lines 30-31 state "This un-conventional design was deployed to alleviate inherent problems of expressing a functional VAR2-CAR fusion-protein in human T cells". The authors should elaborate as to what these inherent problems are.

Response R2-1: Thank you for the comment. The problem was related to getting correct and stable folding of the VAR2 protein portion of the CAR sequence when expressed in human cells. The VAR2 protein only folds correctly in our hands when produced in *E. coli* or in baculovirus. In the revised manuscript (page 6, lines 4-5), we have changed the sentence to read: '*This un-conventional design was deployed to alleviate inherent problems of obtaining correct folding of the VAR2-CAR fusion-protein in human T cells*'.

R2-2: The terminology "conditional CAR" implies that this strategy is tunable using the VAR2-SpyTag protein as an on-off switch. However, the authors are unable to show that injecting the VAR2-SpyTag protein can arm circulating CAR T cells, or discontinuing protein infusion can halt CAR T cell function. A more correct term would be "transient CAR" since that seems to be the effect achieved - Fig 5 shows that multiple doses of CAR T cell injections are required to mediate only modest reduction in tumor growth and prolongation of survival.

Response R2-2: We agree. The term 'Transient CAR' is more accurate. In the revised manuscript, we have changed the title to '*Transient CAR T cells with specificity to oncofetal*'.

glycosaminoglycans in solid tumors'. We have also substituted the word 'conditional' with 'transient' throughout the text.

R2-3: Fig 3B-C: The authors note that the VAR2-targeting CART cells showed more rapid cytotoxicity against the MG-63 cell line than UM-UC-3 despite similar levels of target expression; however this observation has no utility unless they can provide a reason for why this is the case.

Response R2-3: Fair point. Although the observation comes without an obvious explanation, it does indicate that the determinants of CAR T cell efficacy is more complicated than just number of available target molecules. We think this is a conceptually important point to make although we have no experimental evidence to explain it. The reason for the observed discrepancy of efficacy could be related to differences in how the oncofetal CS modifications are presented in the cell lines. The CS glycocalyx is generally dense in tumor cells (see Nature Comms, 2022, PMID: 35963852) and some proteoglycans likely allow for better proximity of the T cells with the cell membrane than other proteoglycans. So, the observed difference may reflect cell line-specific CS proteoglycan repertoires that present oncofetal CS modifications differently. R1 had a related question that we address in R1-5.

R2-4: The specifics of the VAR2-SpyTag protein used in these studies should be more clearly defined, similar to the diagrams of the SpyCatcher CAR in Fig1B-C. Methods state that a minimal CS-binding subunit of the VAR2CSA protein was used, but this should be more immediately obvious to the readers.

Response R2-4: Thank you for the comment. In addition to the description of the VAR2-[SpyC] protein in the result section page 6, lines 10-15, and In the revised manuscript, with we have added more detail in the Method section on **page 17-18, lines 29-31, 1-2.**

Referee #3 (R3):

Khazamipour and colleagues provide highly relevant data on a novel approach to tackle cancer by targeting oncofetal chondroitin sulfate (CS) with a novel CART construct. CS represents an agnostic cancer target and clinical translation of this strategy is of high potential. The authors report for the first time the design and functionality of a conditional CAR T cell product with selectivity to oncofetal CS. The authors 'armed' CARTs with recombinant VAR2CSA lectins (rVAR2) to specifically target cancer cells in in vitro and cancers in murine models in vivo and consequently increased survival. Several minor points should be addressed before a revised version of this excellent work can be accepted in EMBO Molecular Medicine.

Minor points:

R3-1: The authors should comment on potential immunogenicity of VAR2-SpyT. How does this potentially reduce the applicability of this approach in vivo?

Response R3-1: Thank you for this relevant comment. As with any foreign protein sequence, we would expect a degree of immunogenicity to arise over time. However, as evident by the persistence of malaria endemics, it is clear that the VAR2 sequence has been evolutionarily refined by the malaria parasite to have minimal immunogenicity in humans where it is exposed on the surface of infected red blood cells in circulation. Also, in a clinical trial using the VAR2 protein in a vaccine formulation against malaria (Clin Infect Dis., 2019, PMID: 30629148), the protein was only able to produce serum conversion when administered with powerful adjuvants. For the SpyT portion (8 amino acids), there has been no reports on significant immunogenicity in numerous studies using this intein technology in animals. For those reasons, we think it is unlikely that adverse or counter-productive immune reactions would arise in the duration of a potential CAR T

cell treatment. In the revised manuscript, we have discussed this topic on **page 12-13, lines 21-31, 1.**

R3-2: The half-life time of VAR2-SpyT is only 10 minutes, can this be prolonged by modification to improve the arming process in vivo?

Response R3-2: Very interesting question. There are tricks to extend half-life of proteins in vivo including PEG formulations and albumin binding sequences. It is possible that these formulations could improve half-life of the recombinant protein to a point where we would get adequate exposure to un-armed CAR T cells in vivo. However, the risk here is that the formulations may interfere with the SpyT-SpyC covalent interaction. But definitely something that we are considering for future work. In the revised manuscript, we have added a sentence in the discussion highlighting this path as a potential avenue for improvements on **page 12, lines 14-19.**

R3-3: In vivo there is not a 100% efficacy. How does resistance occur or can be explained?

Response R3-3: Thank you for this fundamentally important question. Based on our previous work targeting oncofetal CS glycans with VAR2-drug conjugates (VDC), we see that some tumor types are able to re-grow after VDC treatment. However, if we examine the relapsed tumors for expression of oncofetal CS, they all remain positive for the modification. This indicates that incomplete efficacy of an oncofetal CS-targeting modality is not necessarily related to loss of target expression, but likely rather due to sub-optimal dosing. That said, when we analyze tissue microarrays of different cancer indications for oncofetal CS positivity, then it becomes clear that some indications have small numbers of tumors with low/absent oncofetal CS expression. This means that a tumor can be a tumor without oncofetal CS re-programming. We think that these oncofetal CS low tumors likely have expression of a different and currently unknown glycosaminoglycan subtype that can substitute for oncofetal CS. For that reason, we believe that the most obvious route to resistance will likely be due to compatible glycosaminoglycan substitution, although this hypothesis stands untested. In the revised manuscript, we have discussed this important consideration on **page 11, lines 18-31.**

11th Sep 2024

Dear Dr. Daugaard,

Thank you for submitting your revised study, and please accept my apologies for the unusual delay in getting back to you, which is due the fact that one referee needed more time to provide his/her report, which was then submitted when I was on annual leave.

We have now received the reports from the referees who were asked to re-review your manuscript. As you will see below, while referee #1 is overall satisfied with the revisions, referee #2 still expresses concerns on the use of VAR2-SpyTag as treatment strategy considering the complex manufacturing, multiple injections required, and limited clinical benefit.

After looking at the data again, and upon discussion with my colleagues, we agreed this point should be further discussed, in your rebuttal letter as well as in the manuscript, and the clinical impact should be toned down (including in the abstract and synopsis). Please also discuss the point raised by referee #1.

Additionally, please address the following editorial issues:

1/ Manuscript text:

- Please accept the previous changes and only keep in track changes mode any new modification.
- The following email bounced: mrobert@prostatacentre.com. There is a discrepancy in author's name for Morgan E. Roberts (manuscript) vs Morgan Robert (system). Please check and correct.
- We can accommodate a maximum of 5 keywords, please adjust accordingly.

- Methods:

- o Thank you for providing a reagents and tools table, please remove it from the manuscript and upload it as a separate file.
- o Cells: please indicate whether the cells were authenticated.
- o Animals: please provide the housing and husbandry conditions. Please state details of authority granting ethics approval (IRB or equivalent committee(s), provide reference number for approval. Include a statement of compliance with ethical regulations.
- o Statistics: please provide statements on sample size, blinding, randomization and inclusion/exclusion criteria.
- Data Availability section: please delete the sentence "Expanded View for this article is available online".
- Acknowledgements: The funding listed in this section should match the information entered in the submission system.
- Please rename "Conflict of interest" to "Disclosure statement and competing interests".
- References should be listed alphabetically, with 10 authors before et al, and should be placed before the figure legends.

2/ Figures:

- Please remove the figure titles from the files.
- Figure S5 needs renaming to Figure EV5 in the legends and callouts.
- Please address the queries from our copy editors in the figure legends:
 1. Please note that the exact p values are not provided in the legends of figures 2c-d; 3b; 4c; EV 2c-d; EV 4a.
 2. Please indicate the statistical test used for data analysis in the legends of figures 5b, d.
 3. Please note that in figures 2c-d; 4c; 5b-c; EV 2c-d; there is a mismatch between the annotated p values in the figure legend and the annotated p values in the figure file that should be corrected.
 4. Please note that information related to n is missing in the legend of figure 5c.
 5. Please note that the error bars are not defined in the legend of figure 5c.
 6. Please note that the white arrowheads are not defined in the legend of figure 3a. This needs to be rectified.

3/ Source Data: please include Figure 3A data in the zipped folder for Figure 3 (currently with Figure 2).

4/ Checklist:

- please fill in the section Cell materials/authentication
- please fill in the section Experimental animals/housing and husbandry conditions
- please check that you do not need to fill in the section Core facilities
- please fill in the sections Experiment study design and statistics/blinding and inclusion/exclusion criteria
- please check your entry for the section Data availability/primary datasets deposition (as you did indicate in the manuscript text that no data has been deposited in a public repository).

5/ Thank you for providing a synopsis text. Please remove it from the manuscript and upload it as a separate file.

Please also suggest a visual abstract to illustrate your article as a PNG file 550 px wide and 300-600 px high. A cropped portion of this image will serve as thumbnail for the table of content on our webpage.

6/ As part of the EMBO Publications transparent editorial process initiative (see our Editorial at <http://embomolmed.embopress.org/content/2/9/329>), EMBO Molecular Medicine will publish online a Review Process File (RPF) to accompany accepted manuscripts.

This file will be published in conjunction with your paper and will include the anonymous referee reports, your point-by-point

response and all pertinent correspondence relating to the manuscript. Let us know whether you agree with the publication of the RPF and as here, if you want to remove or not any figures from it prior to publication. Please note that the Authors checklist will be published at the end of the RPF.

I look forward to receiving your revised manuscript.

With kind regards,

Lise Roth

***** Reviewer's comments *****

Referee #1 (Remarks for Author):

The reviewer suggestions have been sufficiently addressed.

Besides the improvement of the presented manuscript, a minor concern should be clarified:

1. UM-UC-3 is an epithelial-like cell isolated from a human urinary bladder carcinoma. The authors used a panel of bladder cancer cell lines "representing both mesenchymal and epithelial origins". Do you expect to have differences in the behaviour of those cell lines to CAR-T cell therapy according to the respective origin/predominant phenotype? Please, add a discussion on this topic.

Referee #2 (Comments on Novelty/Model System for Author):

The experimental data does not support the use of VAR2-SpyTag and SpyCatcher-CART as a treatment strategy.

Referee #2 (Remarks for Author):

My main concern remains that the VAR2-CAR T cells have minimal potency against tumor in vivo (Figure 5). This is in spite of the fact that these require a laborious manufacturing process with separate generation of SpyC-CART and the VAR2-SpyT reagent, ex vivo complexing and then multiple injections, to achieve a survival benefit of a few days. I cannot comprehend any real advantage to this strategy as a treatment for cancer.

2nd revision

Khazamipour et al. 2024

Point-by-point response to reviews

Referee #1 (Remarks for Author):

The reviewer suggestions have been sufficiently addressed.

Besides the improvement of the presented manuscript, a minor concern should be clarified:

R1-1: UM-UC-3 is an epithelial-like cell isolated from a human urinary bladder carcinoma. The authors used a panel of bladder cancer cell lines "representing both mesenchymal and epithelial origins". Do you expect to have differences in the behaviour of those cell lines to CAR-T cell therapy according to the respective origin/predominant phenotype? Please, add a discussion on this topic.

Response R1-1: Thank you for this question. From all the target analysis we have done in bladder cancer and other cancer indications over the years, we have observed a tendency towards mesenchymal tumors having a slightly higher presentation of oncofetal CS than that of epithelial tumors. However, when we performed a dedicated test in lung cancer cells inducing either epithelial-to-mesenchymal transition (EMT) or mesenchymal-to-epithelial transition (MET), we concluded that binding to the rVAR2 reagent was not significantly affected by EMT or MET states, at least in that model system (Nature Comms, 2018, PMID: 30115931). We have included a discussion around this topic on **page 13, lines 14-18**.

Referee #2 (Comments on Novelty/Model System for Author):

The experimental data does not support the use of VAR2-SpyTag and SpyCatcher-CART as a treatment strategy.

Referee #2 (Remarks for Author):

R2-2: My main concern remains that the VAR2-CAR T cells have minimal potency against tumor in vivo (Figure 5). This is in spite of the fact that these require a laborious manufacturing process with separate generation of SpyC-CART and the VAR2-SpyT reagent, ex vivo complexing and then multiple injections, to achieve a survival benefit of a few days. I cannot comprehend any real advantage to this strategy as a treatment for cancer.

Response R2-2: Thank you for the comment. We agree that the SpyC-CART strategy used in this work is not likely to be directly transferrable to the clinic. Rather this work was conducted to establish proof-of-concept for oncofetal CS as a potential target for a CAR T cell approach using the tools at hand. A successful clinical CAR T strategy would likely require a standard CAR expressing a scFv with specificity to oncofetal CS. The scFv's have just been developed by us and VAR2 Pharmaceuticals (Nature Comms, 2024, PMID: 39215044) and will over the next couple of years be tested in various therapeutic formats, including generic CAR T constructs. In fairness to the work presented in the present manuscript, we have reduced the emphasis on clinical translation of the SpyC-CAR T cell strategy, while focusing on PoC for oncofetal CS as a potential target in a CAR T context. These modifications to the text can be found in the Abstract **page 2, line 17**; in the Impact

2nd revision
Khazamipour et al. 2024

Statement **page 3, lines 19-20**; in the Discussion **page 13, lines 1-5**; and in the final summary statement in the Discussion **page 14, lines 13-16**. These modifications are also reflected in the Synopsis.

1st Oct 2024

Dear Dr. Daugaard,

Thank you for submitting your revised files. I am pleased to inform you that your manuscript is accepted for publication and is now being sent to our publisher to be included in the next available issue of EMBO Molecular Medicine!

Please note that in the authors' checklist, I have selected "Yes" in the left column for Experimental animals/housing and husbandry conditions as you have provided this information in the manuscript.

I also cropped a small portion of your synopsis (115px x 70px) to serve as a thumbnail for the table of content (ToC) on our webpage (attached). Changes are usually not allowed at proofing stage, so please let us know immediately if you would rather have a different eToC thumbnail.

With kind regards,

Lise Roth
